# Using firm-level supply chain networks to measure the speed of the energy transition

Johannes Stangl [1], András Borsos[1,2,3] & Stefan Thurner [1,4,5,6] ✉

International climate targets rely on the success of the energy transition, however systematic monitoring on how the economy adopts low-carbon energy remains underdeveloped. Here we use nationwide supply-chain network data to reconstruct energy portfolios for 25,000 Hungarian firms between 2020 and 2024, covering 75% of gas, 70% of electricity, and 50% of oil consumption. This allows us to quantify the speed of the energy transition -the transition towards low-carbon electricity- on the firm level. We find substantial heterogeneity in decarbonization progress: half of firms increase low-carbon energy shares, but 50% reduce it. Energy cost structures are closely associated with transition behavior, indicating technology-related lock-in effects. Extrapolating current trends yields an aggregate low-carbon share of 20% by 2050, highlighting ineffective decarbonization efforts. If firms strictly adopted strategies of decarbonization frontrunners within their industry sectors, a low-carbon share of 70% could be achieved by 2050, putting climate targets within reach.

The global transition from fossil fuels to low-carbon energy sources is crucial for achieving international climate goals[1]. The energy transition has two major components: decarbonizing the electricity sector and electrifying appliances and industrial processes[2,3]. For some industries and processes, alternative strategies such as biomass, hydrogen, or carbon capture and storage or carbon capture and usage might be more viable options for greenhouse gas mitigation[4–7]. However, there is widespread agreement that direct electrification is the primary pathway for most end-use sectors and processes[3,8–10]. There is an extensive body of literature that analyzes the decarbonization of the electricity sector across different scales—global, regional, and national—using energy system models to project the rollout of low-carbon energy technologies[11–14]. Recently, advances in electricity sector decarbonization have been observed in countries and regions around the world, driven by falling prices for renewable energy technologies such as photovoltaic (PV) installations, wind power, and grid-scale batteries[15–17]. Although many options for the electrification of industrial processes have been discussed[18–21], comprehensive studies on the actual adoption of these technologies remain scarce or focused on specific industries[22].

The literature on industrial electrification can be divided into two dominant streams. The first stream consists of firm-level case studies and small-scale empirical analyses, typically focused on energy-intensive industries subject to emissions trading schemes or other regulatory policies[22–25]. Although case studies provide insights into firm-specific behaviors, barriers, and opportunities, they generally lack generalizability and therefore offer only fragmented evidence on how firms approach electrification. Some studies have used larger panels of firm- or plant-level data to investigate the role of electrification for energy efficiency or productivity, but a focus on the broader energy transition is lacking[26,27]. The second stream examines sector-level patterns of energy consumption and electrification options from a top-down perspective[28–30]. Sector-level research is typically enabled by readily available input-output tables, which are often augmented with environmental data to form environmentally extended input-output tables. These frameworks enable the analysis of sectoral energy consumption and electrification patterns[31,32]. However, this coarse sectoral view limits our understanding of the underlying dynamics, as firms within the same industry sector do not necessarily use the same technologies or follow the same adoption strategies[33,34]. Emerging

---

[1]Complexity Science Hub, Vienna, Austria. [2]National Bank of Hungary, Budapest, Hungary. [3]Institute for New Economic Thinking, Oxford, UK. [4]Supply Chain Intelligence Institute Austria, Vienna, Austria. [5]Medical University of Vienna, Vienna, Austria. [6]Santa Fe Institute, Santa Fe, NM, USA. ✉e-mail: stefan.thurner@meduniwien.ac.at

research on firm-level supply chain networks highlights significant heterogeneity between firms, even within fine-grained industry classifications, including differences in input-output structures[35], exposure to climate transition risks[36], and systemic relevance in the context of climate policy[37]. The Hungarian supply chain network, in particular, has been studied with respect to structural characteristics[38], the systemic relevance of individual firms[39], the amplification of financial systemic risk through supply-chain disruptions[40], and its temporal dynamics[41]. Its characteristics are also extensively discussed in a recent review of firm-level supply chain network research[42].

Here we show how firm-level supply chain network data can be used to reconstruct the energy portfolios of a large and representative sample of firms within a country, enabling an extensive firm-level analysis of energy transition dynamics. We thereby bridge top-down sector-level analyses and bottom-up studies of individual firms, providing a comprehensive firm-level perspective on the energy transition. Firm-level supply chain network data based on value-added tax (VAT) or electronic invoicing records are increasingly available across a growing number of jurisdictions, enabling the application of the approach developed here to monitor firm-level energy transition dynamics in other countries and regions with comparable data[43]. As an empirical test case, we use semi-annual snapshots of the Hungarian supply chain network derived from VAT records between 2020 and 2024 to reconstruct the annual energy consumption portfolios of a sample of 25,231 firms. We identify energy-providing firms—suppliers of electricity, gas, and oil products—based on NACE 4-digit classifications[44] as well as energy-consuming firms as illustrated in Fig. 1a, b. The sample comprises firms with continuous time series of electricity, gas, and oil consumption, revenue, and employment from 2020 to 2024 (see "Methods"). This sample covers 75% of gas, 70% of electricity, and 50% of oil consumption relative to the total firm population; the uncovered share corresponds to firms with inconsistent and incomplete time series and is excluded to ensure robustness (see "Methods" and Supplementary Discussion). By applying semi-annual energy prices, we convert monetary transactions in the supply chain network into kilowatt-hours of energy consumption. We determine low-carbon electricity consumption, $L_i(t)$, by multiplying each firm's electricity consumption, $E_i(t)$, by the low-carbon share of Hungary's electricity mix, $u(t)$. Nuclear energy is included as a source of low-carbon electricity, in line with its classification by both the Intergovernmental Panel on Climate Change (IPCC) and Hungary's 90% low-carbon electricity target by 2030[3,45]. We compute a firm's low-carbon share, $l_i(t)$, as the ratio of $L_i(t)$ to its total energy consumption, $T_i(t)$ (Fig. 1c). To quantify firm-level transition dynamics, we estimate two indicators. We fit a linear function to $l_i(t)$ to obtain the decarbonization trend, $\delta_i$, and an exponential function to obtain the decarbonization rate, $\lambda_i$ (Fig. 1d). These capture two technological change patterns: an incremental mode (linear) that assumes steady and gradual progress, and a disruptive mode (exponential) that reflects more rapid transitions, such as those driven by capital stock renewal, both documented in prior technological adoption studies[46,47]. Using these measures, we address three questions: How heterogeneous are firms within and across sectors in their adoption of low-carbon electricity? What characteristics distinguish transitioning firms —those with positive $\delta_i$ and $\lambda_i$—from non-transitioning firms? And, are current firm-level trends sufficient to achieve an energy transition in line with international climate targets?

## Results
### Firm-level heterogeneity of energy consumption
The analysis of firm-level energy portfolios shows significant heterogeneity in both low-carbon electricity consumption and decarbonization trends across industry sectors, as shown in Fig. 2. The scatter plot in Fig. 2a shows average low-carbon electricity consumption, $\overline{L}_i$, on the $y$-axis (logarithmic scale) versus the decarbonization trend, $\delta_i$, on the $x$-

axis (linear scale) for every firm $i$. Every marker represents one firm, colors indicate its NACE 1-digit industry category, and the size represents its average revenue $\overline{R}_i$. We exclude firms from the NACE category 'D - Electricity, gas, steam and air conditioning supply' that act as energy providers, as our analysis focuses on the energy consumption of end users; see Method section for details. A complete list of the NACE 1-digit industry category codes and their descriptions can be found in Supplementary Table 1. The plot shows a wide range of decarbonization trends, with firms of all sizes, low-carbon electricity consumption levels, and sector affiliations exhibiting both positive and negative decarbonization trends, $\delta_i$. The majority of firms show decarbonization trends centered around zero, meaning that their low-carbon share, $l_i$, remained relatively stable over the observation period. The top consumers of low-carbon electricity are primarily found in the sector 'C - Manufacturing', along with firms in the sector 'G - Wholesale and retail trade; repair of motor vehicles and motorcycles'. The plot also highlights considerable variability in decarbonization trends across both sectors and individual firms. While low-carbon electricity consumption and decarbonization trends span nearly the entire space, a considerable gap exists in the bottom-right corner of Fig. 2a, where small electricity consumers with high decarbonization rates are practically absent. This suggests that predominantly larger firms and those with higher electricity consumption dominate the progress in increasing their low-carbon share. The majority of small consumers (defined as those with low-carbon consumption, $\overline{L}_i < 10^3$ kWh) exhibit negative decarbonization trends, with 1,308 firms showing a decline of their low-carbon share compared to 762 firms displaying positive trends. Figure 2b shows the distribution of the low-carbon share, $\overline{l}_i$, for the NACE 2-digit industry sectors. While median values are generally low, notable within-sector heterogeneity exists, along with variation between NACE 2-digit sectors within the same NACE 1-digit category. Manufacturing sectors, which are among the largest consumers of low-carbon electricity (as shown in Fig. 2a), exhibit higher median low-carbon shares. Service sectors such as 'J - Information and communication' and 'M - Professional, scientific, and technical activities', located higher up in the NACE industry classification, also show higher median values. Across all sectors, some firms approach a low-carbon share of 0.7, indicating near-exclusive use of low-carbon electricity. The upper bound for the low-carbon share is 0.7378, representing the proportion of low-carbon electricity sources in the Hungarian electricity mix in 2024 (see "Methods" section for more details). Although some firms have made significant progress in increasing their low-carbon share, the majority still show relatively modest levels.

### Characteristics of transitioning firms
To determine which firms tend to transition, Fig. 3 shows the relation of several firm characteristics with their transition status: transitioning, ($\delta_i > 0$ and $\lambda_i > 0$), and not transitioning ($\delta_i < 0$ or $\lambda_i < 0$). More specifically, Fig. 3 presents a forest plot of the Adjusted Odds Ratios (AOR) from a multivariate logistic regression analysis for the following firm characteristics for each NACE 1-digit sector: (a) average fossil energy cost share of revenue, $\overline{fc}_i$, (b) average electricity cost share of revenue, $\overline{ec}_i$, (c) average revenue, $\overline{R}_i$, (d) average employment, $\overline{em}_i$, and (e) average total energy consumption, $\overline{T}_i$. Note that we exclude sector 'B - Mining and Quarrying' and 'O - Public administration and defense; compulsory social security' from the regression analysis because too few observations were available to estimate the coefficients. The AOR represents the change in odds of transitioning for a 10% increase in the respective firm characteristic. Figure 3a shows that a higher fossil cost share is associated with a significantly lower likelihood of transitioning across NACE 1-digit sectors. In contrast, a higher electricity cost share is associated with a greater likelihood of transitioning, as shown in Fig. 3b. Larger firms in terms of revenue tend to exhibit a lower likelihood of transitioning, as shown in Fig. 3c, whereas higher total energy

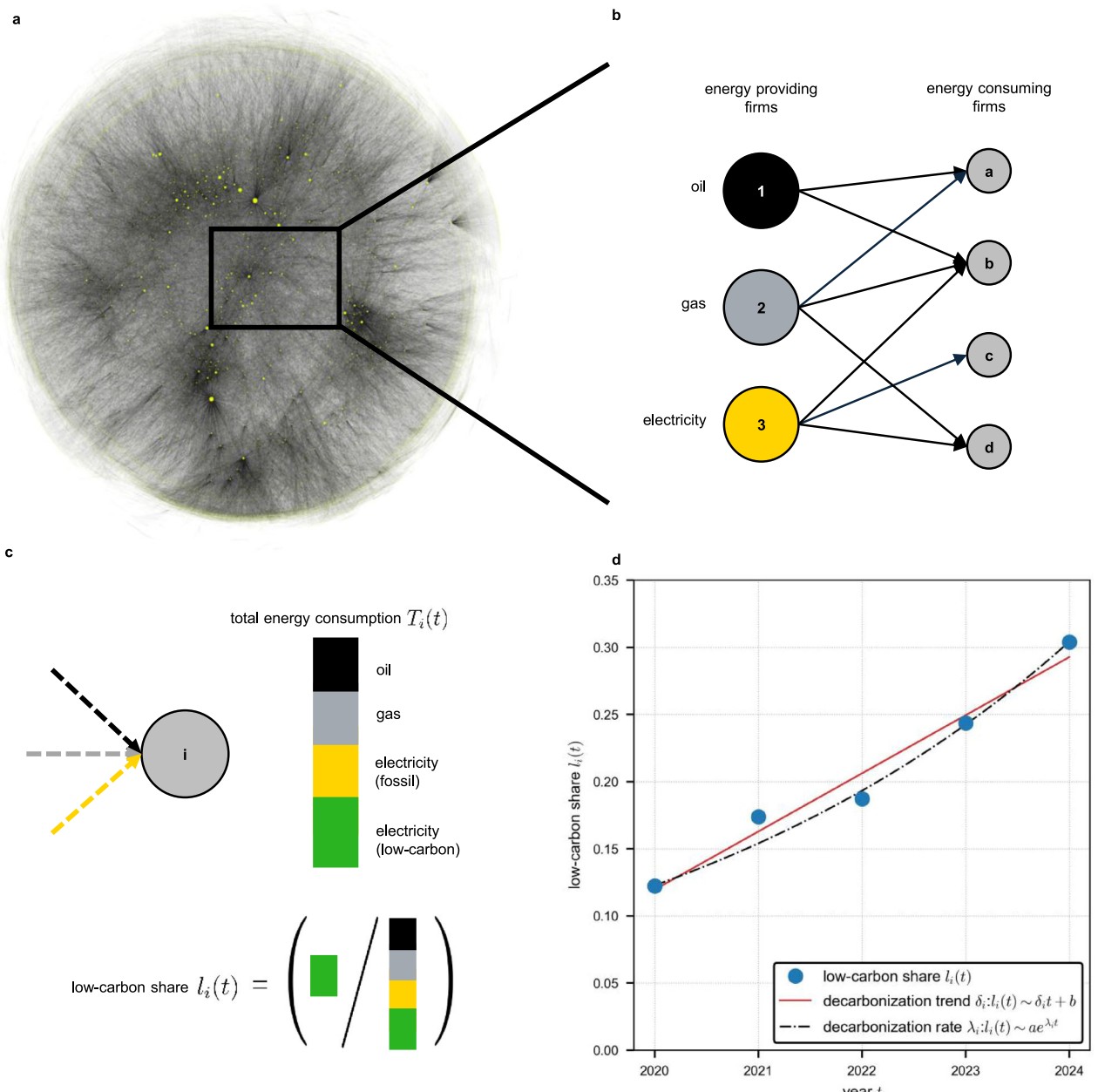

**Fig. 1 | Measuring the speed of the energy transition by reconstructing the energy mix of individual firms. a** Hungarian supply chain network aggregated over six months. Nodes represent firms and links represent supply relations; node size corresponds to each firm's out-strength. **b** Schematic micro-level view of the network distinguishing between energy-providing firms (left) and energy-consuming firms (right). Energy providers supply electricity, gas, and oil products to consuming firms. For each firm, energy consumption by carrier is obtained by converting VAT-based monetary transactions into kilowatt-hours using semi-annual energy prices. **c** Energy consumption portfolio of firm $i$ in year $t$. Low-carbon electricity consumption, $L_i(t)$, equals electricity consumption, $E_i(t)$, multiplied by the low-carbon share of Hungary's electricity mix, $u(t)$ (see "Methods"). The firm's low-carbon share, $l_i(t)$, is defined as the ratio of $L_i(t)$ to total energy consumption, $T_i(t)$. **d** Low-carbon share, $l_i(t)$, of firm $i$ over the observation period. The pace of the transition toward low-carbon energy use is quantified by the decarbonization trend, $\delta_i$ (linear regression), and the decarbonization rate, $\lambda_i$ (exponential fit).

consumption is associated with a greater likelihood of transitioning, as shown in Fig. 3e. The number of employees does not appear to have a significant association with the likelihood of transitioning across most NACE 1-digit sectors, as shown in Fig. 3d. Table 1 reports the estimated adjusted odds ratios together with their 95% confidence intervals. The associations appear strongest for the fossil cost share, electricity cost share and total consumption. The adjusted odds ratios for a 10% increase in fossil energy cost share show a significant decrease in the likelihood of transitioning for most sectors, supporting the hypothesis that high fossil energy costs are associated with non-transitioning firms. The adjusted odds ratios for a 10% increase in electricity cost

share show a consistent increase in the likelihood of transitioning, supporting the hypothesis that higher electricity costs are more common among transitioning firms. While higher overall consumption seems to be associated with a higher likelihood of transitioning away from fossil fuels, higher revenue seems to decrease the likelihood of being a transitioning firm. This may indicate that firms with higher revenues can afford to continue paying rising fossil energy prices rather than making potentially risky investments to change their capital stock. These results are significant across most NACE 1-digit sectors and indicate that a firm's energy cost structure plays a central role in determining whether it transitions to low-carbon electricity

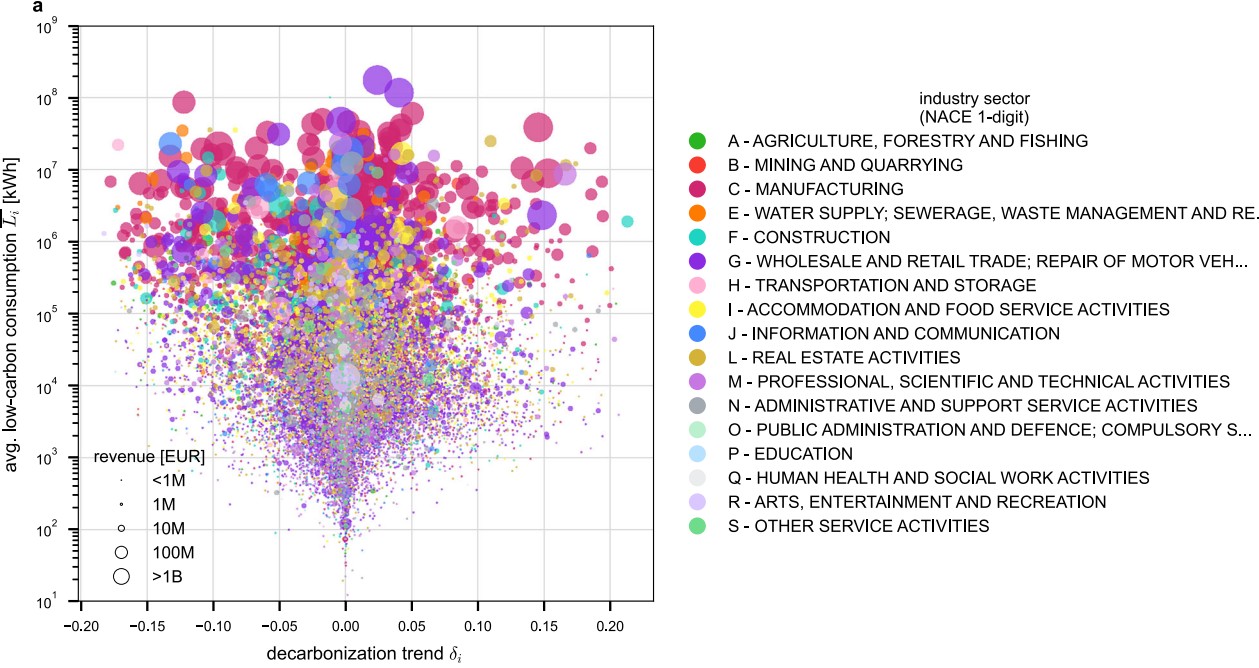

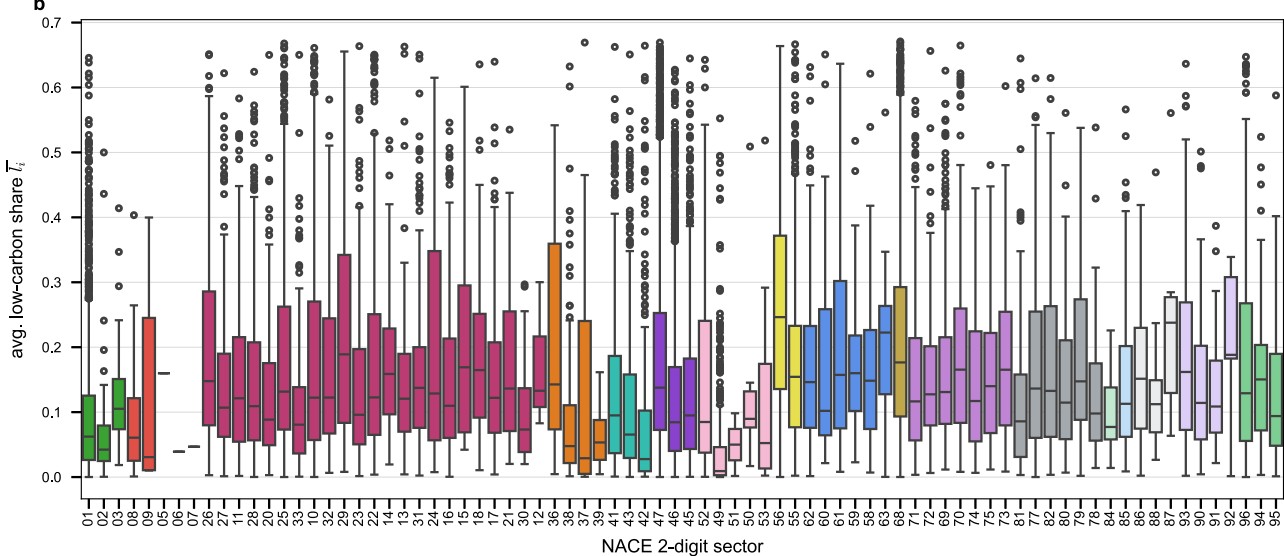

**Fig. 2 | Heterogeneity of low-carbon electricity consumption and decarbonization trends at the firm level. a** Scatter plot of firms' average low-carbon electricity consumption, $\overline{L_i}$, versus their decarbonization trend, $\delta_i$. Each point represents one firm, colored by its NACE 1-digit industry sector; marker size is proportional to the firm's average annual revenue. **b** Box plots of firms' average low-carbon electricity share, $\overline{l_i}$, grouped by NACE 2-digit industry sectors. Each box summarizes the distribution of $\overline{l_i}$ across firms within a sector. The center line indicates the median; the box bounds correspond to the 25th and 75th percentiles (interquartile range, IQR); whiskers extend to the most extreme data points within $1.5 \times$ IQR from the quartiles; points beyond this range are shown as outliers. The number of firms (sample size $n$) contributing to each sectoral box plot is reported as exact values in Supplementary Table 2. No control group was used, as the analysis is observational and compares naturally occurring variation across firms and industry sectors.

consumption. The observed higher electricity cost shares for transitioning firms and higher fossil energy cost shares for non-transitioning firms could be explained by a 'lock-in effect', where firms are constrained by their current energy technologies. The upfront investments required to switch to low-carbon electricity may further deter firms from transitioning.

**Energy transition scenarios**

To examine whether the current pace of the energy transition aligns with international climate goals, Fig. 4 presents scenarios of future aggregated energy consumption for all firms in the sample. Specifically, it displays four scenarios illustrating how the energy share between fossil electricity (yellow), low-carbon electricity (green) and fossil energy (black) may evolve until 2050, based on the observed decarbonization trends, $\delta_i$, and decarbonization rates, $\lambda_i$; see "Methods" section for details. All panels show the aggregated fossil share, $f(t)$, fossil electricity share, $e_f(t)$, and low-carbon share, $l(t)$, of all firms in the sample during the period 2020–2024 as solid area plots. For reference, Hungary's historical shares, as derived from final energy consumption tables, are displayed for the years 2014–2019 as shaded area plots with reduced opacity. While not directly comparable to the low-carbon shares estimated from the aggregated firm sample, these

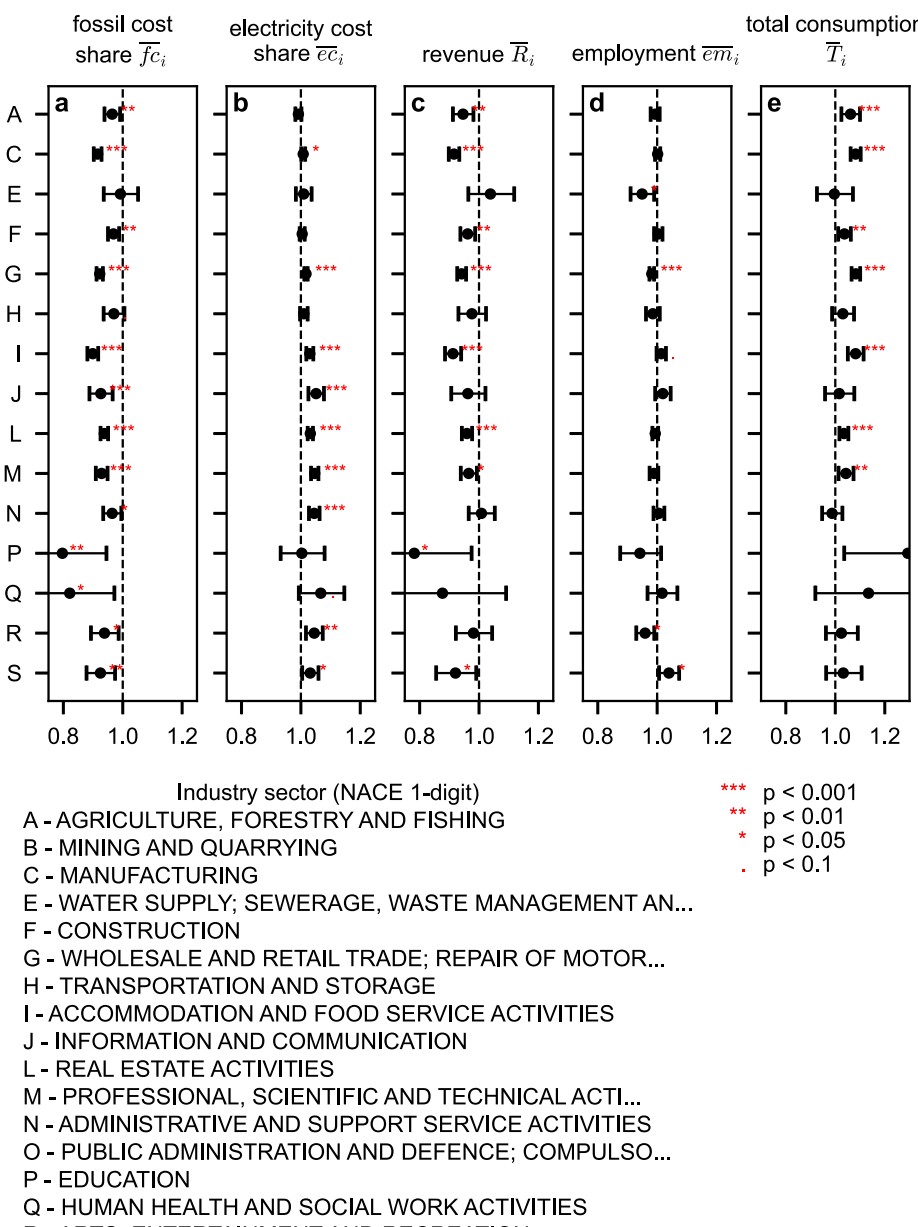

**Fig. 3 | Characteristics of transitioning and non-transitioning firms.** Forest plots showing adjusted odds ratios (AORs) from multivariate logistic regression models estimating the association between firm transition status (transitioning firms: $\delta_i > 0$ and $\lambda_i > 0$; non-transitioning firms: $\delta_i < 0$ or $\lambda_i < 0$) and firm-level characteristics, separately for each NACE 1-digit industry sector. Points indicate the estimated AOR associated with a 10% increase in the corresponding log-transformed firm characteristic, computed as $e^{\beta \cdot 0.1}$, where $\beta$ denotes the estimated logistic regression coefficient. Horizontal error bars indicate the corresponding 95% confidence intervals. Vertical dashed lines indicate odds ratios of 1 (no association). Logistic regressions were estimated by maximum likelihood; statistical significance was assessed using two-sided Wald tests. No adjustment was made for multiple comparisons. Superscript symbols denote significance levels (*** $p < 0.001$, ** $p < 0.01$, * $p < 0.05$,. $p < 0.1$). The corresponding numerical results for the AOR, 95% confidence intervals, and exact $p$-values are reported in Table 1. Firm characteristics include **a** average fossil cost share, $\overline{fc}_i$, **b** average electricity cost share, $\overline{ec}_i$, **c** average revenue, $\overline{R}_i$, **d** average employment, $\overline{em}_i$, and **e** average total energy consumption, $\overline{T}_i$. Sectors 'B - Mining and Quarrying' and 'O - Public administration and defense; compulsory social security' were excluded due to insufficient sample size for reliable estimation.

historical values provide a context for Hungary's energy transition progress. Forecasted aggregated low-carbon shares, $l(t)$, and corresponding fossil shares, $f(t)$, are depicted as hatched area plots. For the forecasted values, we assume that firms continue to consume their average total energy, $\overline{T}_i$ as observed for the period 2020–2024. The electricity mix of Hungary is assumed to gradually decarbonize until the early 2030s, aligning with Hungary's goal of achieving 90% low-carbon electricity by 2030. For details, see "Methods" and Supplementary Methods 2. Figure 4a depicts the business-as-usual scenario,

based on observed linear decarbonization trends, $\delta_i$. Firms with positive trends increase their share of low-carbon electricity, while firms with negative trends reduce it over time. The plot shows that the system quickly saturates, with $l(t)$ reaching only 0.198 by 2050; see Table 2. This suggests that no substantial transition occurs if the current linear trends persist. Figure 4b illustrates the business-as-usual scenario, based on measured exponential decarbonization rates, $\lambda_i$. Firms with positive rates significantly increase their share of low-carbon electricity, while those with negative rates decrease it over

**Table 1 | Adjusted odds ratios (AORs) from multivariate logistic regressions for each NACE 1-digit industry sector analyzing the association between firm transition status and firm characteristics averaged over the observation period**

| Sector | fossil cost share $\overline{fc}_i$ | | electricity cost share $\overline{ec}_i$ | | revenue $\overline{R}_i$ | | employment $\overline{em}_i$ | | total consumption $\overline{T}_i$ | |
|---|---|---|---|---|---|---|---|---|---|---|
| | AOR | 95% CI | AOR | 95% CI | AOR | 95% CI | AOR | 95% CI | AOR | 95% CI |
| A | 0.964** (0.0093) | 0.939, 0.991 | 0.992 (0.1096) | 0.981, 1.002 | 0.946** (0.0029) | 0.912, 0.981 | 0.993 (0.3808) | 0.978, 1.009 | 1.062** (0.0009) | 1.025, 1.100 |
| C | 0.915*** (<0.0001) | 0.902, 0.929 | 1.008 (0.0201) | 1.001, 1.014 | 0.916*** (<0.0001) | 0.898, 0.934 | 1.002 (0.761) | 0.993, 1.010 | 1.082*** (<0.0001) | 1.062, 1.103 |
| E | 0.992 (0.7900) | 0.936, 1.052 | 1.009 (0.0488) | 0.983, 1.036 | 1.038 (0.3208) | 0.964, 1.118 | 0.949 (0.0146) | 0.910, 0.990 | 0.996 (0.9140) | 0.926, 1.071 |
| F | 0.969** (0.0013) | 0.950, 0.988 | 1.004 (0.3103) | 0.996, 1.013 | 0.962** (0.0031) | 0.937, 0.987 | 1.004 (0.6087) | 0.990, 1.018 | 1.037* (0.0039) | 1.012, 1.063 |
| G | 0.922*** (<0.0001) | 0.912, 0.933 | 1.017*** (<0.0001) | 1.012, 1.023 | 0.941*** (<0.0001) | 0.927, 0.956 | 0.982*** (<0.0001) | 0.974, 0.990 | 1.083*** (<0.0001) | 1.066, 1.101 |
| H | 0.970 (0.0977) | 0.936, 1.006 | 1.010 (0.1284) | 0.997, 1.023 | 0.976 (0.3176) | 0.931, 1.024 | 0.985 (0.2192) | 0.962, 1.009 | 1.030 (0.1714) | 0.987, 1.076 |
| I | 0.899*** (<0.0001) | 0.881, 0.917 | 1.030*** (<0.0001) | 1.018, 1.042 | 0.912*** (<0.0001) | 0.885, 0.940 | 1.014 (0.0876) | 0.998, 1.030 | 1.082*** (<0.0001) | 1.051, 1.114 |
| J | 0.926*** (0.0004) | 0.888, 0.966 | 1.051*** (<0.0001) | 1.025, 1.078 | 0.962 (0.2086) | 0.906, 1.022 | 1.019 (0.1487) | 0.993, 1.046 | 1.016 (0.5928) | 0.959, 1.077 |
| L | 0.938*** (<0.0001) | 0.925, 0.951 | 1.032*** (<0.0001) | 1.023, 1.041 | 0.959*** (<0.0001) | 0.942, 0.977 | 0.994 (0.1983) | 0.984, 1.003 | 1.035*** (0.0001) | 1.017, 1.053 |
| M | 0.929*** (<0.0001) | 0.909, 0.949 | 1.047*** (<0.0001) | 1.034, 1.059 | 0.965 (0.0137) | 0.939, 0.993 | 0.989 (0.1475) | 0.975, 1.004 | 1.043*** (0.0046) | 1.013, 1.074 |
| N | 0.964* (0.0242) | 0.934, 0.995 | 1.045*** (<0.0001) | 1.027, 1.063 | 1.008 (0.7097) | 0.965, 1.053 | 1.006 (0.5306) | 0.988, 1.024 | 0.987 (0.5339) | 0.947, 1.029 |
| P | 0.797* (0.0091) | 0.671, 0.945 | 1.003 (0.9327) | 0.932, 1.080 | 0.782 (0.0290) | 0.627, 0.975 | 0.942 (0.1084) | 0.876, 1.013 | 1.293* (0.0230) | 1.036, 1.614 |
| Q | 0.821* (0.0218) | 0.694, 0.972 | 1.067 (0.0767) | 0.993, 1.146 | 0.877 (0.2390) | 0.705, 1.091 | 1.017 (0.5125) | 0.968, 1.068 | 1.134 (0.2397) | 0.919, 1.399 |
| R | 0.938* (0.0121) | 0.893, 0.986 | 1.045** (0.0014) | 1.017, 1.074 | 0.981 (0.5506) | 0.922, 1.044 | 0.959* (0.0103) | 0.930, 0.990 | 1.025 (0.4489) | 0.962, 1.091 |
| S | 0.925** (0.0032) | 0.878, 0.974 | 1.031* (0.0225) | 1.004, 1.059 | 0.921* (0.0270) | 0.856, 0.991 | 1.040* (0.0190) | 1.006, 1.074 | 1.032 (0.3703) | 0.963, 1.106 |

For each characteristic and sector, the table reports the AOR associated with a 10% increase in the corresponding log-transformed firm characteristic, computed as $e^{0.1\beta}$ and the accompanying 95% confidence interval (95% CI). Exact $p$-values are reported in parentheses in the AOR column (to four decimal places, or as $p < 0.0001$ when smaller), and statistical significance is additionally indicated by superscript symbols (*** $p < 0.001$, ** $p < 0.01$, * $p < 0.05$, $\cdot$ $p < 0.1$). Statistical significance was assessed using two-sided Wald tests. No adjustment was made for multiple comparisons. Firm characteristics include average fossil cost share, $\overline{fc}_i$, average electricity cost share, $\overline{ec}_i$, average revenue, $\overline{R}_i$, average employment, $\overline{em}_i$, and average total energy consumption, $\overline{T}_i$. Sectors 'B - Mining and Quarrying' and 'O - Public administration and defense; compulsory social security' were excluded due to insufficient sample size for reliable estimation.

time. Initially, the overall low-carbon share, $l(t)$, rises but plateaus around 2030, as most firms with positive rates fully decarbonize by then. By 2050, the low-carbon share reaches a value of only 0.266. This indicates that even with exponential adoption of electrification options, the system-wide transition to low-carbon electricity would fall short of meeting international climate goals. Firms currently increasing their fossil energy shares must reverse this trend, as explored in the transition scenarios in Fig. 4c, d. Figure 4c shows a transition scenario, where every firm with a negative linear decarbonization trend, $\delta_i$, adopts a positive trend derived from an industry peer within the same NACE 4-digit sector, matched by average revenue and number of employees. This means that every firm adopts the strategies of a decarbonization frontrunner within its industry, resulting in all firms having positive decarbonization trends, $\delta_i$ from $t = 2024$ onward. In this optimistic scenario, the forecasted low-carbon share, $l(t)$, rises sharply until the mid-2030s, as Hungary's electricity mix becomes fully decarbonized. Beyond that point, $l(t)$ continues to grow steadily, reaching a share of 0.548 by 2050; see Table 2. However, even with industry-wide adoption of positive linear decarbonization trends, $\delta_i$, the energy transition on the firm level remains insufficient to achieve the rapid emission reductions required to meet climate targets, i.e., net-zero emissions by 2050[1]. Figure 4d presents the 'best-case' scenario, where every firm with a negative exponential decarbonization rate, $\lambda_i$, adopts a positive rate from $t = 2024$ onward from a decarbonization frontrunner within the same NACE 4-digit sector, matched by average revenue and number of employees. This scenario assumes that all firms adopt the best-case decarbonization strategies of their peers, resulting in universal positive decarbonization rates. Here, the forecasted low-carbon share, $l(t)$, increases rapidly until the early 2030s, when Hungary's electricity mix is forecasted to be fully decarbonized. Beyond this point, $l(t)$ continues to climb steeply, reaching a share of 0.699 by 2050. This suggests that in a best-case scenario, where all firms pivot decisively towards decarbonization, the energy transition at the firm level could become consistent with achieving the Paris climate goals, especially given that electrification is typically associated with significant gains in energy efficiency[3]. Remaining fossil fuel consumption could additionally be compensated by other means than direct electrification, such as biomass, hydrogen and synthetic fuels, or carbon capture and storage/usage, which are not accounted for in this analysis. We provide an uncertainty analysis of the presented energy scenarios in the Supplementary Discussion and Supplementary Fig. 13.

## Discussion

Achieving climate neutrality in line with international targets is critically dependent on the success of the energy transition, much of which must be borne by industry and the wider economy. Although existing research has largely concentrated on the decarbonization of the electricity sector, comprehensive evidence on how firms themselves are adopting low-carbon electricity remains scarce. Here, we use detailed firm-level supply-chain network data to reconstruct the energy portfolios of 25,231 energy-consuming firms in Hungary. This approach enables us to measure the speed of the energy transition at the firm level, uncovering patterns that remain hidden in aggregate sector-level statistics.

We find substantial heterogeneity within and across industry sectors in low-carbon electricity consumption, the share of low-carbon electricity in the overall energy mix, and the decarbonization speeds of firms. Although roughly half of the firms exhibit positive decarbonization trends, an evenly large share has reduced its low-carbon electricity share. This indicates that the energy transition is far from assured and that a significant share of firms continues to rely on fossil energy sources. While heterogeneity across sectors is expected and widely documented, the pronounced heterogeneity within sectors suggests that technological feasibility alone does not explain observed

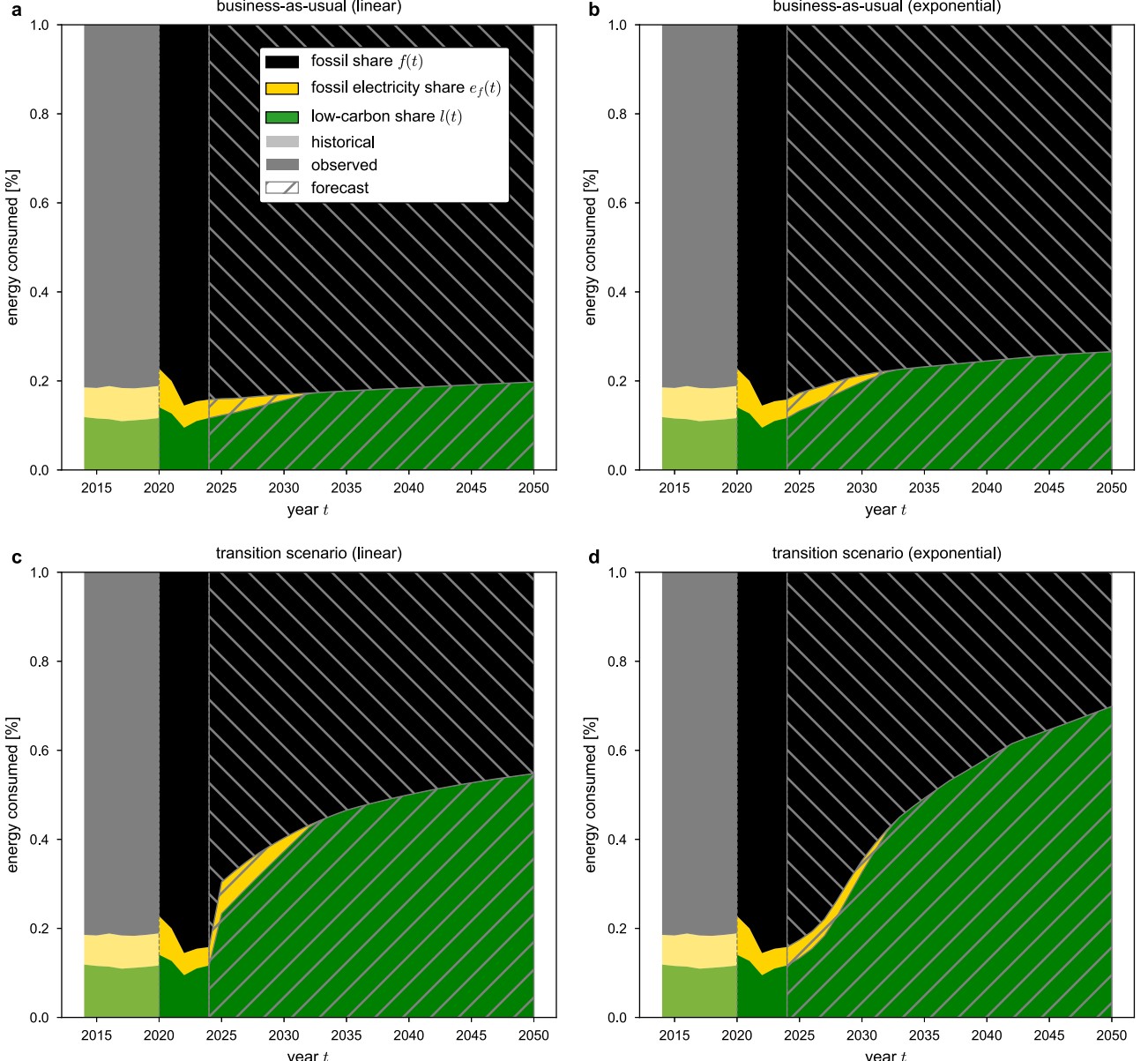

**Fig. 4 | Scenarios of future energy consumption from firm-level decarbonization trajectories.** Each panel shows aggregate relative energy consumption decomposed into fossil share $f(t)$, fossil electricity share $e_f(t)$, and low-carbon share $l(t)$ over the period 2014–2050. Here, decarbonization trends and rates are determined by the interaction between firms' electrification dynamics and forecasts of the low-carbon share of Hungary's electricity mix, $u(t)$ (see "Methods" and Supplementary Methods 2. **a** business-as-usual (linear): Observed firm-level decarbonization trends, $\delta_i$, are extrapolated for all firms. **b** business-as-usual (exponential): Observed firm-level decarbonization rates, $\lambda_i$, are extrapolated for all firms. **c** transition scenario (linear): Firms with negative decarbonization trends are

assigned positive decarbonization trends from the closest firm within the same NACE 4-digit sector, defined by similarity in revenue and number of employees (see "Methods"). Future aggregate low-carbon and fossil shares are then calculated based on these adjusted trends. **d** transition scenario (exponential): Firms with negative decarbonization rates, $\lambda_i$, are assigned positive decarbonization rates from the closest firm within the same NACE 4-digit sector, defined by similarity in revenue and number of employees (see "Methods"). Future aggregate low-carbon and fossil shares are then calculated according to these adjusted rates. Table 2 reports forecasted low-carbon and fossil shares for 2020, 2030, 2040, and 2050 across all scenarios.

transition patterns. Instead, firm-specific conditions, such as existing capital stock, cost pressures, managerial choices, and regulatory exposure, appear to play a central role in shaping divergent transition trajectories among firms engaged in similar economic activities. The firm-level approach developed in this study makes it possible to examine these differences in detail and to better understand heterogeneous transition behavior.

We find that transitioning and non-transitioning firms display distinct and consistent characteristics across most NACE 1-digit industry sectors. Transitioning firms are more likely to exhibit higher electricity cost shares, while non-transitioning firms tend to have

higher fossil cost shares. These results suggest that energy cost structures play a central role in shaping transition behavior. Our regression analysis further indicates that greater total energy consumption is associated with a higher likelihood of transitioning, which may suggest that more energy-intensive firms face stronger pressures to reduce fossil dependence. By contrast, higher revenues are associated with a lower likelihood of transitioning, suggesting that larger firms may prefer to absorb rising fossil fuel costs rather than pursue potentially risky investments in new capital stock. Employment levels, in turn, show little association with the likelihood of transitioning. Taken together, these patterns are consistent with what has been

**Table 2 | Forecasts for low-carbon share, $l(t)$, and fossil share, $f(t)$, of aggregated energy consumption for the years 2020, 2030, 2040, and 2050 under the linear and exponential business-as-usual scenarios and the linear and exponential transition scenarios, as shown in Fig. 4**

| year | business-as-usual (linear) | | business-as-usual (exponential) | | transition (linear) | | transition (exponential) | |
|------|--------|--------|--------|--------|--------|--------|--------|--------|
| | $l(t)$ | $f(t)$ | $l(t)$ | $f(t)$ | $l(t)$ | $f(t)$ | $l(t)$ | $f(t)$ |
| 2020 | 0.141 | 0.772 | 0.141 | 0.772 | 0.141 | 0.772 | 0.141 | 0.772 |
| 2030 | 0.157 | 0.830 | 0.197 | 0.787 | 0.373 | 0.597 | 0.326 | 0.647 |
| 2040 | 0.185 | 0.815 | 0.245 | 0.755 | 0.500 | 0.500 | 0.582 | 0.418 |
| 2050 | 0.198 | 0.802 | 0.266 | 0.734 | 0.548 | 0.452 | 0.699 | 0.301 |

Note that values $l(t)$ and $f(t)$ for 2020 are the same, since all scenarios start from the same observed low-carbon and fossil shares.

described in the literature as a 'lock-in effect,' whereby existing energy technologies constrain firms' flexibility, making fossil-reliant firms less likely to transition away from fossil fuels and electricity-reliant firms more likely to further adopt low-carbon energy sources. The substantial upfront investments required to replace fossil-based capital may reinforce such dynamics, potentially slowing the transition even when long-term economic or environmental incentives exist. This investment barrier has also been highlighted in previous studies[3,9,10].

While our findings document significant correlations between firms' energy cost structures, size, and transition behavior, the empirical design does not allow causal inference, and the following policy implications should therefore be interpreted with caution. Future research is needed to identify the underlying mechanisms using quasi-experimental or other causal inference approaches (see Supplementary Discussion). Nevertheless, our findings point to several potential avenues for policy interventions. One option is lowering the investment barriers that could prevent firms from transitioning away from fossil-based technologies. Targeted support, such as subsidies for electrification options (e.g., heat pumps) or state-backed loans with favorable conditions, can help firms manage the high upfront costs of replacing existing capital. Once these investments are made, it is plausible and supported by evidence from our study that firms continue their low-carbon paths, reinforcing progress over time. At the same time, the finding that high-revenue firms are less likely to transition could point to the need for instruments that make continued fossil fuel use less attractive. Stronger carbon pricing is one broad option. In addition, since our study provides a tool to identify frontrunners and laggards within fine-grained industry segments, policymakers could design more targeted measures, such as sectoral emission standards or performance-based incentives that reward greener firms through mechanisms like tax breaks or preferential access to public procurement. Encouragingly, we find a positive correlation between total energy consumption and transition behavior, which is particularly important given the leverage of large energy consumers in reducing emissions. Firms with high energy demand that demonstrate a credible decarbonization pathway could likewise benefit from favorable treatment.

To assess whether current trends align with international climate goals, we simulate a set of energy scenarios based on observed decarbonization speeds and potential shifts in firms' strategies. These scenarios are not intended as precise forecasts, but rather to illustrate how observed firm-level behavior aggregates into long-run transition pathways. The projections point to a substantial gap between present dynamics and climate goals. Under a continuation of current trends, low-carbon shares reach only 20–26% by 2050, indicating that existing transition dynamics are unlikely to deliver rapid aggregate decarbonization. This gap is driven by the continued expansion of fossil energy use among lagging firms, which offsets progress achieved by frontrunners, suggesting that the energy transition is characterized by polarization rather than gradual convergence. The widespread presence of frontrunner firms across fine-grained NACE 4-digit sectors

indicates that technological feasibility is not the primary constraint on the transition; instead, uneven diffusion, strategic delay, and technological uncertainty appear to play a central role. To explore the untapped potential of this within-sector heterogeneity, we construct two counterfactual transition scenarios in which firms adopt the decarbonization trends and rates of their industry's decarbonization frontrunners. These scenarios illustrate how a decisive shift in firms' strategies could substantially alter aggregate outcomes, raising low-carbon shares to 55% under best-practice linear decarbonization trends and to 70% under exponential decarbonization rates by 2050. The contrast between business-as-usual and counterfactual transition pathways suggests that accelerating the energy transition depends less on technological breakthroughs than on mechanisms that promote diffusion, reduce adjustment frictions, and discourage strategic delay, underscoring that the transition is fundamentally a coordination and incentive problem.

Our study faces several data-related limitations. The analysis is restricted to the period from 2020 to 2024. Prior to 2020, the quality of firm-level data was less reliable due to changes in the thresholds for reporting VAT transactions. Specifically, firms were required to report VAT transactions only when they reached a certain transaction value threshold, which decreased over time. After 2020, this threshold was removed entirely, as described in ref. 42. Only from this point onward does a comprehensive reconstruction of firms' energy portfolios become possible. Although 2020-2024 allows one to obtain trends in the firm-level energy transition, it is a relatively short time span. This window, therefore, captures a snapshot of recent transition dynamics rather than long-run adjustment processes. Nevertheless, this period is particularly relevant, as electrification options and renewable energy technologies have become more affordable, and climate policies, such as those in the EU Green Deal, introduced in late 2019, have gained increasing momentum among policymakers and industry. In this respect, the 2020-2024 period represents the onset phase of firm-level decarbonization.

Our study faces another data-related limitation concerning the conversion of energy expenditures into kilowatt-hours using EURO-STAT price data stratified by consumption bands. This approach masks heterogeneity in effective prices within each band, arising from differences in load profiles, contractual arrangements, on-site generation, and access to renewable procurement. For instance, organizations with predominantly daytime loads, such as universities, logistics hubs, or service-sector firms, may face lower effective electricity prices due to greater opportunities for solar self-generation, while energy-intensive industrial firms operating around the clock may face higher effective prices. These differences also imply varying opportunities for decarbonization, since daytime-oriented consumers are better positioned to integrate self-generation or time-sensitive procurement options. As a result, low-carbon electricity shares may be underestimated for daytime-oriented firms and overestimated for industrial firms with flatter load profiles. If this bias is systematic, the general trends in our fitted models are likely to be less affected,

although the intercept may shift. Therefore, aggregate low-carbon shares in the scenario analysis may be somewhat overstated, given the high energy use of industrial consumers. Future work could incorporate more granular pricing data or load-profile information to better capture these firm-level cost differences.

Another important data constraint is that we do not observe energy generation activities carried out directly on firms' premises, such as photovoltaic (PV) installations. Our analysis is restricted to electricity purchases from other companies and therefore omits behind-the-meter PV generation that firms self-consume. Calculations provided in the Supplementary Discussion indicate that commercially self-consumed PV electricity increased from roughly 0.6 TWh in 2020 to about 2.5 TWh in 2024, corresponding to around 27% of total national PV output and covering an estimated 2.3% of total commercial electricity demand in 2020 and about 8.8% in 2024. Although this growth is substantial, the overall magnitude remains modest relative to total commercial consumption, particularly for large industrial users. The impact is therefore likely heterogeneous across sectors, with comparatively larger effects for service-sector firms than for large industrial consumers. As a consequence, our derived low-carbon electricity shares for large industrial consumers are likely to be less affected. By contrast, the low-carbon electricity shares $l_i(t)$ for smaller service-sector firms may be understated. This may not hold for firms that have invested in large-scale PV systems dedicated to self-consumption[48], but because we cannot identify these firms, this remains a limitation of our analysis. Future work could address this limitation by integrating data on solar PV installations if such information becomes available.

Beyond data constraints, our analysis also faces methodological limitations. Oil products are highly heterogeneous, ranging from fuels such as gasoline and diesel to feedstocks like naphtha. Since we do not have product-level information but only observe monetary transactions in the supply chains between firms, we must make assumptions about which oil products firms consume in order to convert monetary values into kilowatt-hours via prices. Specifically, we assumed that firms consume oil in the form of diesel and gasoline and applied the weighted average price of these fuels in Hungary to convert expenditures into kilowatt-hours. This provides a reasonable estimate of the energy consumed through oil products, as diesel and gasoline are by far the most widely consumed oil products in Hungary, with the exception of the chemical industries, where naphtha is the most prevalent. For more details, see the "Methods" and Supplementary Methods 3.

Another methodological limitation is that fluctuations in energy prices, especially for electricity and gas during the energy crisis, are only partly captured in our data. In 2022, for example, we observe a decline in the aggregate low-carbon share in the scenarios shown in Fig. 4. This decrease may be linked to the sharp rise in electricity prices during the crisis, which appears to be only partly reflected in the expenditure patterns firms in our firm sample. While our sample reasonably reflects relative oil and gas consumption, it appears to underrepresent electricity use and overrepresent gas use to some extent relative to official statistics. This might be because firms procure electricity from additional, unobserved sources such as own production or the spot market (see Supplementary Discussion). Overall, our methodology is more reliable in periods of relatively stable energy prices, as indicated by the stable aggregate shares for 2023 and 2024. Further details on the incorporation of energy price data are provided in the Methods and Supplementary Methods 3.

A further methodological limitation is that our focus is on electrification as a pathway to low-carbon energy, which is viable for most sectors but less applicable in industries such as chemicals or cement, where alternative strategies are needed. Future research is needed to examine these alternatives if detailed product-level data on firm-to-firm transactions becomes available, enabling a broader assessment of decarbonization pathways beyond electrification and a more comprehensive understanding of firm-level transitions.

Furthermore, we focus on direct energy consumption, but future research could extend our analysis by examining how supplier-customer relationships influence firm-level transition behavior. An exploratory assessment suggests that firms' transition dynamics are more closely aligned with those of their customers than with those of their suppliers (see Supplementary Discussion).

The study also faces limitations related to the sample construction. Energy providers typically sell both gas and electricity, but are categorized under only one NACE code, either for the distribution of gas or electricity. To address this issue, we limited our analysis to firms with connections to both gas and electricity providers, on the basis that firms with connections to both gas and electricity providers are most likely to purchase these energy types from separate providers. This restriction resulted in a sample of 25,231 firms, which still accounts for a substantial share of total energy consumption in Hungary (25% of total final energy consumption in 2023, including transport and residential consumption: 16.6% of oil, 40.0% of gas, and 17.1% of electricity), making it a reasonable representation of the broader firm population. Further details on this selection process are found in the "Methods" section. A detailed discussion of the resulting firm sample is presented in the Supplementary Discussion.

Finally, the generalizability of our findings beyond Hungary should be treated with caution. Certain features of the Hungarian economy—such as its continued reliance on fossil fuels, the presence of energy-intensive industries, the expansion of nuclear power, and the rapid uptake of solar PV—are also characteristic of other Central and Eastern European countries. While the quantitative results remain context-specific, some qualitative insights may extend beyond Hungary, including the structural role of energy costs in shaping transition behavior and the finding that nearly all fine-grained industry sectors contain both frontrunners rapidly adopting low-carbon electricity and laggards continuing to expand fossil fuel use.

To conclude, our analysis indicates that across all industrial sectors, frontrunners and laggards coexist even within fine-grained NACE 4-digit classifications, grouping firms with highly similar activities. This suggests that the adoption of low-carbon electricity is shaped not by technological feasibility alone but also by firm-specific constraints and strategic choices. At the same time, the continued expansion of fossil energy use among a substantial share of firms points to persistent structural frictions that may slow aggregate decarbonization. By leveraging firm-level supply chain network data, this study develops a framework for analyzing decarbonization dynamics at the micro level. As such data become increasingly available across regions[43], this approach can be used to monitor national energy transition progress and to inform more targeted and differentiated policy measures to accelerate the transition.

## Methods

### Reconstructing annual firm energy use from the Hungarian supply chain network

This study relies on the reconstructed firm-level supply chain network of the Hungarian economy, derived from VAT transaction data collected by the Hungarian National Tax and Customs Administration. Since 2014, this data, made available through the Central Bank of Hungary, has enabled the identification of supply relationships between firms, facilitating the creation of detailed snapshots of the Hungarian supply chain network at the firm level. More information on this dataset can be found in previous studies[38,39]. The dataset anonymizes the companies but provides information such as revenue, number of employees, and industry sector, enabling analysis based on company size and activities. The coverage of the dataset has evolved over time due to changes in reporting thresholds for VAT transactions. Between early 2015 and mid-2018, only transactions with a cumulative

tax content exceeding 1 million Hungarian Forint (HUF) within a reporting period (monthly, quarterly, or annually) were recorded. From the third quarter of 2018 to mid-2020, the threshold was lowered to 100,000 HUF and applied to individual transactions, significantly increasing the visibility of firms and their supply relationships. However, firms whose transactions consistently fell below this new threshold were excluded. Since the third quarter of 2020, all inter-firm invoices must be reported, eliminating thresholds entirely and providing a comprehensive view of supply chain relationships. This is also discussed in detail in ref. 42. Larger firms in Hungary are required to report their VAT on a monthly basis, which allows a fine-grained resolution of the monetary transactions between firms. Only smaller firms are required to report on an annual basis.

## Identifying energy providers

For this study, we use semi-annual snapshots of the Hungarian firm-level supply chain network from 2020 to 2024, as this period offers the most consistent and comprehensive time-series data following the removal of reporting thresholds. To identify energy providers within the network, we categorize firms based on their NACE 4-digit industry affiliations. Electricity providers are identified as firms classified under one or more of the following categories: 'D35.1 - Electric power generation, transmission, and distribution,' 'D35.1.1 - Production of electricity,' 'D35.1.2 - Transmission of electricity,' 'D35.1.3 - Distribution of electricity,' and 'D35.1.4 - Trade of electricity.' Gas providers are identified as firms in the categories 'D35.2.1 - Manufacture of gas,' 'D35.2.2 - Distribution of gaseous fuels through mains,' and 'D35.2.3 - Trade of gas through mains.' Oil providers are identified as firms classified under 'B6.1.0 - Extraction of crude petroleum,' 'C19.2.0 - Manufacture of refined petroleum products,' 'G47.3.0 - Retail sale of automotive fuel in specialized stores,' and 'G46.7.1 - Wholesale of solid, liquid, and gaseous fuels and related products.' It is important to note that multiple NACE 4-digit categories are included for each type of energy provider (electricity, gas, and oil) to account for the limitations of industry classifications. Firms' NACE classifications do not always capture the full scope of their activities. For instance, a firm might simultaneously transport and sell gas or refine and sell oil products, yet only be classified under a single category. Moreover, energy companies often operate as large entities consisting of multiple sub-companies with different industry classifications. To ensure comprehensive coverage of energy consumption, we include all relevant categories associated with these activities. Note that coal usage is not covered in this study due to the absence of a distinct NACE code specifically related to coal distribution. Additionally, coal in Hungary is primarily used for electricity production, which is already addressed in the electricity mix of Hungary. Beyond that, coal consumption is limited to a small number of firms, mostly in the steel and pulp and paper industries[49].

## Firm sample construction

We aggregate the monetary inputs from companies identified as providers of electricity, gas, and oil products—based on the industry classifications detailed above—on a semi-annual basis over the observation period from 2020 to 2024. This aggregation enables us to estimate the total amount of electricity, gas, and oil purchased by each firm semi-annually, allowing us to track their energy consumption trends over time. Firms with an annual reporting requirement are kept in the sample. To ensure data quality, we apply several restrictions to our sample. Since energy providers often supply both gas and electricity, we retain only firms that simultaneously purchase electricity and gas from companies classified under electricity and gas provider categories, based on their NACE 4-digit sector affiliation, for each year of the observation period. This ensures we account for firms with distinct electricity and gas purchases. Firms classified under any NACE 4-digit industry related to energy provision—whether electricity, gas,

or oil—are excluded to focus on end-users of energy, rather than energy suppliers. Additionally, firms from the financial sector, classified under the NACE 1-digit category 'K—Financial and insurance activities' are excluded, as they may act as energy brokers without directly consuming energy themselves. Furthermore, firms from the NACE 4-digit category 'H52.2.1—Service activities incidental to land transportation' are excluded from the analysis, as firms from this category might be involved in liquefaction of gas for transportation purposes, which likely means their purchases are not linked to the consumption of gas directly. We exclude firms participating in the European Union Emissions Trading System (ETS), as their energy costs are affected by additional expenditures for carbon credits, which cannot be accounted for in our analysis of firm characteristics. Our dataset is anonymized, allowing us to identify whether a company participates in the ETS but not to determine which specific firms are involved. Estimating their energy-related carbon credit costs would require individual treatment that is not feasible within the scope of this study. For this reason, we adopt a conservative approach and exclude ETS firms from the sample, amounting to 201 firms in total. Firms without a NACE classification in the dataset, and those lacking revenue data for any observed year, are also excluded. We further exclude an outlier firm whose gas consumption increased anomalously by three orders of magnitude from one year to the next. After applying these criteria, the final sample consists of 25,231 firms for which continuous semi-annual time-series data on electricity, gas, oil consumption, and revenue are available. Although this filtering approach may exclude firms that exclusively use gas or electricity, it is necessary to maintain the consistency and reliability of the resulting firm sample. Since energy providers often supply both gas and electricity, and we cannot distinguish between these energy types in the monetary transactions within our dataset, the assumption of separate providers for electricity and gas ensures the robustness of the analysis. Details on the firm sample coverage and its comparison with sectoral energy consumption data from Hungary's energy balance[50] is provided in the Supplementary Discussion, Supplementary Fig. 7 and Supplementary Fig. 8.

## Conversion of monetary inputs into energy consumption via energy prices

We use semi-annual energy prices to convert the monetary inputs for electricity, gas, and oil into kilowatt-hours of energy consumed by each firm in the sample. For electricity and gas, we rely on energy price data for non-household users in Hungary from EUROSTAT, which includes all taxes and levies[51,52]. This data is collected from electricity and gas providers who report the prices paid by their customers across different consumption bands (seven for electricity and six for gas). We determine the appropriate consumption band for each firm by converting the energy consumption bands' thresholds into monetary units using the respective energy prices. We then assign each firm to the corresponding band for each half-year based on its expenditures. For firms with an annual VAT reporting requirement, we derive annual energy prices by taking the average of the semi-annual energy prices. This applies to between 1,000 and 5,000 firms in any given year in our sample. After applying the semi-annual energy prices, we aggregate oil, gas, and electricity consumption to the annual level for further processing to avoid seasonality effects. Supplementary Fig. 2 and Supplementary Fig. 3 show the price trends for electricity and gas. Over the observation period, electricity and gas prices have fluctuated significantly, particularly during the energy crisis of 2022, when prices surged. They subsequently decreased again in 2023. It is also important to note that price evolution varies across different consumer groups: larger firms typically experienced earlier price increases and reductions, while smaller firms saw these changes later. This discrepancy may be attributed to larger firms hedging their energy costs through the energy futures market, while smaller firms often have fixed contracts with one or more energy providers.

**Government support during the energy crisis.** In 2022, the Hungarian government introduced a support scheme aimed at energy-intensive small- and medium-sized enterprises (SMEs), covering 50% of the increased electricity and gas costs. This scheme impacted an estimated 10,000 companies[53]. In 2023, the government also implemented a price cap on electricity, benefiting around 5000 companies in sectors such as manufacturing, accommodation, and warehousing/transport[54]. To our best knowledge, these market interventions are unlikely to affect our energy consumption estimates, as the SME support scheme functioned as a reimbursement, and the electricity price cap applied directly to the energy bills of companies. Consequently, the average prices reported by EUROSTAT still represent the actual prices paid by companies to their energy providers and, therefore, the corresponding kilowatt-hours consumed.

## Estimating oil consumption based on fuel prices

To estimate oil consumption, we assume that firms primarily use oil in the form of fuels and employ fuel price trends as a proxy to convert observed oil expenditures into energy units. Data from the National Detailed Energy Balance, provided by the Hungarian Energy and Public Utility Regulatory Authority (MEKH), confirms that diesel and gasoline are the dominant forms of oil product consumption[50]. Supplementary Fig. 4 shows the distribution of oil product consumption for 2023, derived from final consumption data in the National Detailed Energy Balance. Diesel accounts for nearly half of oil product consumption in 2023, followed by gasoline at 20%. Naphtha represents 14% of total consumption. However, according to the National Detailed Energy Balance, the chemical and petrochemical industries are the sole consumers of naphtha. Therefore, the assumption that firms primarily consume oil in the form of fuels is reasonable for all industry sectors, except for the chemical industry, which also consumes oil in the form of naphtha. Since naphtha is generally cheaper than diesel or gasoline, our method may underestimate oil product consumption in the chemical sector. Fuel price data is obtained from historical data of the Weekly Oil Bulletin provided by the European Commission[55]. To determine a single representative price for oil products, we calculate a weighted average of gasoline and diesel prices in Hungary, using weights based on the relative consumption of these fuels, also derived from the Weekly Oil Bulletin data. This approach implicitly assumes that firms consume these fuels in similar proportions. On average, diesel represents 74% of fuel consumption, while gasoline accounts for 26% during the observation period from 2020 to 2024. This weighted average price allows us to convert monetary expenditures on oil products into kilowatt-hours consumed. Supplementary Fig. 5 depicts the evolution of fuel prices for gasoline, diesel, and their weighted average in Hungary from 2018 to 2024. Fuel prices have risen significantly since 2021, reflecting the broader energy crisis in Europe.

## Calculating the low-carbon share $l_i(t)$

To determine the low-carbon electricity consumption, $L_i(t)$, for a firm $i$ in a given year, we use the low-carbon share of Hungary's annual electricity mix, as provided by the online platform Ember[56]. Ember reports annual data on clean and fossil electricity generation in terawatt-hours, excluding imports and exports. We compute the low-carbon share of Hungary's annual electricity mix, $u(t)$, by dividing the electricity generated from low-carbon sources (including hydro, PV, wind, bioenergy, and nuclear) by the total electricity produced. The low-carbon electricity consumption of a firm in a given year is calculated as

$$L_i(t) = E_i(t) \cdot u(t),  \quad (1)$$

where $E_i(t)$ represents the total electricity consumption of firm $i$. The low-carbon share of a firm's total energy consumption is then given by

$$l_i(t) = \frac{L_i(t)}{T_i(t)},  \quad (2)$$

where $T_i(t)$ denotes the total energy consumption of firm $i$ in year $t$. This approach allows us to estimate the low-carbon share, $l_i(t)$, for each firm in our sample over the period from 2020 to 2024, enabling us to analyze trends and assess the pace at which firms are increasing the low-carbon share of their energy mix. We therefore assume that all firms source from the same electricity mix and thus do not account for individual efforts to procure clean electricity. However, given that Hungary's electricity market is highly concentrated, with one firm group holding a leading position in the market, this assumption may be considered reasonable for the average firm sourcing from the most prevalent provider (see also ref. 49).

## Measuring the speed of the energy transition at the firm level

To measure the pace of the energy transition for individual firms, we use two distinct models of technology adoption. The first approach assumes that the decarbonization process follows a gradual, steady trend, while the second approach models the transition as a potentially faster, more disruptive process.

In the first model, we fit a linear function to the low-carbon share, $l_i(t)$, for each firm $i$ over the observation period. This allows us to calculate the decarbonization trend, $\delta_i$, which represents the gradual pace of decarbonization. The linear equation is

$$l_i(t) = \alpha_i + \delta_i \cdot t,  \quad (3)$$

where $\alpha_i$ is the initial low-carbon share and $\delta_i$ is the slope of the trend.

Instead of ordinary least squares (OLS), we employ a robust regression estimator based on the Huber loss function[57], which downweights the influence of large residuals by combining quadratic and linear loss.

$$\rho(r) = \begin{cases} \frac{1}{2} r^2 & \text{if } |r| \leq k \\ k|r| - \frac{1}{2} k^2 & \text{if } |r| > k \end{cases}  \quad (4)$$

for residuals, $r = l_i(t) - \alpha_i - \delta_i \cdot t$, where $k$ is a tuning constant. We set $k$ to the default value used in the Python `statsmodels` implementation, $k = 1.345$[58]. Small residuals are treated quadratically (as in OLS), while large residuals receive linear weighting, reducing the influence of outliers. The optimal values of $\delta_i$ and $\alpha_i$ are thus obtained by solving

$$(\widehat{\alpha}_i, \widehat{\delta}_i) = \arg\min_{\alpha_i, \delta_i} \sum_{t=2020}^{2024} \rho\left(l_i(t) - \alpha_i - \delta_i \cdot t\right).  \quad (5)$$

The second model assumes that the low-carbon share, $l_i(t)$ follows an exponential trajectory, with decarbonization characterized by a potentially faster growth process:

$$l_i(t) = \beta_i \cdot e^{\lambda_i \cdot t},  \quad (6)$$

where $\beta_i > 0$ is the initial level and $\lambda_i$ is the exponential growth rate.

For estimation, we take logarithms, which transforms the exponential specification into a linear model in log-space:

$$ln\left(l_i(t)\right) = \gamma_i + \lambda_i \cdot t,  \quad (7)$$

where $\gamma_i = ln(\beta_i)$. This allows us to estimate $\lambda_i$ as the slope of a regression of $ln(l_i(t))$ on time. We employ the same robust regression procedure as in the linear model, now applied to the log-transformed

data. For residuals

$$r = ln(l_i(t)) - \gamma_i - \lambda_i \cdot t , \qquad (8)$$

the optimal values $(\hat{\gamma}_i, \hat{\lambda}_i)$ are obtained by solving

$$(\hat{\gamma}_i, \hat{\lambda}_i) = \arg\min_{\gamma_i, \lambda_i} \sum_{t=2020}^{2024} \rho \left( ln(l_i(t)) - \gamma_i - \lambda_i \cdot t \right) , \qquad (9)$$

where $\rho(\cdot)$ denotes the Huber loss function defined above. The original parameter of interest $\beta_i$ can then be recovered as $\hat{\beta}_i = e^{\hat{\gamma}_i}$. Supplementary Methods 4 and Supplementary Fig. 6 demonstrate that robust estimation provides more reliable results than OLS by reducing the influence of outliers in selected example firms.

While logistic growth models are also commonly used in studies of technological adoption, we refrain from using them here. This is because logistic growth requires fitting a three-parameter model, and with only five time points, 2020–2024, such a model would not be reliable or meaningful for this analysis. Instead, the linear and exponential models are better suited for illustrating two distinct patterns of technological change: steady, incremental transitions and more rapid, disruptive shifts.

## Quantifying the relationship between firm characteristics and transition status

We perform multivariate logistic regression analyses to examine the relationship between firm characteristics and their transition status within each NACE 1-digit industry sector. Transitioning firms are defined as those with a positive decarbonization trend, $\delta_i > 0$, and a positive exponential decarbonization rate, $\lambda_i > 0$, while non-transitioning firms have negative values for either of the two indicators. Ambiguous cases, where only one of the two indicators is negative, are conservatively also classified as non-transitioning. The binary outcome variable is coded as 1 for transitioning firms and 0 otherwise.

As predictors, we use averaged firm-level variables over the observation period, applying logarithmic transformations to mitigate the influence of extreme values or skewness. The predictor set combines indicators of energy costs and firm size: average fossil energy cost share, $\overline{fc}_i$, average electricity cost share, $\overline{ec}_i$, average firm revenue, $\overline{R}_i$, average number of employees, $\overline{em}_i$, and average total energy consumption, $\overline{T}_i$.

For each NACE 1-digit sector, we estimate a logistic regression of the form

$$\text{logit}(P(\text{trans}_i)) = \alpha + \sum_j \beta_j X_{ij} , \qquad (10)$$

where $P(\text{trans}_i)$ is the probability that firm $i$ transitions, and $X_{ij}$ denotes the log-transformed predictor variables. The regression coefficients, $\beta_j$, capture how each firm characteristic relates to the probability of transitioning, conditional on the others.

To aid interpretation, we compute odds ratios associated with a 10% increase in each predictor:

$$\text{OR}_{1\%} = e^{\beta_j \cdot 0.1} , \qquad (11)$$

with corresponding confidence intervals based on standard errors. An odds ratio greater than 1 indicates that an increase in the predictor raises the odds of transitioning, while a value below 1 suggests the opposite.

## Decarbonization of Hungary's electricity sector

To create scenarios for future low-carbon electricity consumption, we need to make a forecast of the evolution of the low-carbon share, $u(t)$, of Hungary's national electricity mix. This share influences the low-

carbon share, $l_i(t)$, for each firm $i$, as it reflects the overall decarbonization progress of the country's electricity sector. Between 2020 and 2024, Hungary's low-carbon share grew significantly, from about 60% in 2020 to over 70% in 2024. This increase was mainly due to the expansion of photovoltaic (PV) installations. To forecast $u(t)$ for 2025 to 2050, we use a linear regression based on the observed trend from 2020 to 2024. The resulting forecast for $u(t)$, is consistent with Hungary's target of achieving 90% low-carbon electricity by 2030. This would require an annual increase of 3.2 in the share of low-carbon electricity. Under this forecast, Hungary's electricity sector will be fully decarbonized by 2033 and remain so thereafter. Supplementary Fig. 1 and Supplementary Table 3 provide further details on the observed values of $u(t)$ and the forecasted path for $u(t)$.

## Forecasting future low-carbon shares $l_{i,\text{ forecast}}(t)$

To forecast the future low-carbon share, $l_i(t)$, for each firm $i$, we first need to make an assumption regarding future total energy consumption, $T_i(t)$. We assume that each firm's energy consumption will remain at the average level observed between 2020 and 2024 $\overline{T}_i = \frac{1}{5}\sum_{t=2020}^{2024} T_i(t)$.

This assumption is conservative because many electricity-powered appliances and processes are more efficient than their fuel-based counterparts[3]. However, the potential overestimation of energy use is counterbalanced by the possibility of firm growth and capital expansion, which may increase energy consumption.

It is important to note that the observed decarbonization trends, $\delta_i$, and decarbonization rates, $\lambda_i$, of firms cannot be directly used to forecast their future low-carbon shares because $l_i(t)$ already reflects the change of Hungary's low-carbon share, $u(t)$. Therefore, we perform separate robust regressions to estimate a linear electrification trend, $\epsilon_i$, and an exponential electrification rate, $\mu_i$ for each firm $i$, analogous to the regressions in equations (5) and (9). These regressions are based on the share of electricity in firms' energy mixes, excluding Hungary's low-carbon share $e_i(t) = \frac{E_i(t)}{T_i(t)}$.

Depending on the scenario—whether a linear or exponential adoption is assumed—the forecasted low-carbon shares for each firm $i$ in the years 2025 to 2050 are calculated using one of the following equations

$$l_i(t) = (\alpha_i + \epsilon_i \cdot t) \cdot u(t) , \qquad (12)$$

or

$$l_i(t) = (\beta_i \cdot e^{\mu_i \cdot t}) \cdot u(t) . \qquad (13)$$

In both equations $\alpha_i$ and $\beta_i$ represent the initial values for each firm's electrification trend or rate. The forecasted fossil share is then calculated as

$$f_i(t) = 1 - l_i(t) . \qquad (14)$$

## Modeling the linear business-as-usual scenario

In the linear business-as-usual scenario, firms continue their current electrification trends, $\epsilon_i$, while Hungary's electricity mix gradually decarbonizes. The forecasted low-carbon share for each firm $i$ in year $t$ is calculated according to equation (12). To project the total low-carbon and fossil energy shares consumed by all firms from 2025 to 2050, we first compute the low-carbon and fossil energy consumption for each firm $i$ in each year $t$

$$L_i(t) = l_i(t) \cdot \overline{T}_i , \qquad (15)$$

$$F_i(t) = f_i(t) \cdot \overline{T}_i , \qquad (16)$$

where $\overline{T_i}$ represents the firm's total energy consumption, assumed constant over time. The total low-carbon share of energy consumption across all firms is then given by

$$l(t) = \sum_{i=1}^{n} \frac{L_i(t)}{\sum_{j=1}^{n} \overline{T_j}} , \qquad (17)$$

where $n$ is the number of firms. Finally, the total fossil energy share is determined as $f(t) = 1 - l(t)$.

### Modeling the exponential business-as-usual scenario
In the exponential business-as-usual scenario, firms continue to follow their current electrification rates, $\mu_i$, while Hungary's electricity mix gradually decarbonizes. The forecasted low-carbon share for each firm $i$ in year $t$ is computed using equation (13). The total low-carbon and fossil shares of energy consumption across all firms are then determined using the same methodology as detailed above.

### Constructing the linear transition scenario
In the linear transition scenario, all firms are required to decarbonize by adopting a positive electrification trend, $\epsilon_i$. For firms that do not exhibit a positive electrification trend, a counterfactual trend is assigned based on a matched firm $j$ with similar characteristics. The matching is conducted by first restricting the candidate pool to firms operating within the same NACE 4-digit sector as firm $i$. From this pool, the firm $j$ that most closely matches firm $i$ in terms of average revenue, $\overline{R_i}$, and average number of employees, $\overline{em_i}$, is selected using a nearest-neighbor procedure. Revenue data are available for the period 2020–2023, and employment data for 2020-2022. If no firm with a positive electrification trend is available within the same NACE 4-digit sector, the matching procedure is expanded to the corresponding NACE 2-digit sector, and the same nearest-neighbor criteria are applied. Through this procedure, each firm is assigned a positive decarbonization trend, $\mu_i$. The forecast low-carbon share for firm $i$ in year $t$ is then computed using equation (12), and total low-carbon and fossil shares are calculated as described above.

### Constructing the exponential transition scenario
In the exponential transition scenario, all firms are required to decarbonize rapidly by adopting a positive electrification rate, $\mu_i$. For any firm $i$ that does not follow a positive electrification rate, it is paired with a firm $j$ exhibiting the most similar characteristics. The pairing process follows the same procedure outlined earlier. As a result, every firm is assigned a positive electrification rate, $\mu_i$. The forecasted low-carbon share for each firm $i$ in year $t$ is then calculated according to equation (13), and the total low-carbon and fossil shares are computed in the same manner as described above.

### Software used
Data collection and analysis were performed using R and Python computing environments. Data collection and preprocessing were conducted in R (v4.2.2) using RStudio (v2023.06.1) and the package `data.table` (v1.14.8), `igraph` (v1.5.0) and `Matrix` (v1.5.1), as well as in Python (v3.12.3) using Jupyter Server (v2.14.1) and Visual Studio Code (v1.108.0) with the packages `numpy` (v1.26.4) and `pandas` (v2.2.2). Data analysis and visualization were performed in the same R environment and in the Python environment using `numpy` (v1.26.4), `pandas` (v2.2.2), `matplotlib` (v3.9.0), `plotly` (v5.22.0), `scikit-learn` (v1.5.2), `statsmodels` (v0.14.4) and `seaborn` (v0.13.2).

### Reporting summary
Further information on research design is available in the Nature Portfolio Reporting Summary linked to this article.

## Data availability
The raw data on financial transactions between Hungarian value-added tax-paying firms are protected and are not available due to data privacy laws. Requests for collaborations to work with these data can be addressed to olahzs@mnb.hu.

## Code availability
The code[59] supporting the analyses, including the aggregation of purchases related to energy inputs, the estimation of decarbonization trends and rates, the multivariate logistic regressions of firm characteristics, and the construction of the energy scenarios, is available at https://github.com/jo-stangl/using_firm-level_supply_chain_networks_to_measure_the_speed_of_the_energy_transition.

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

## Acknowledgements

This work was supported in part by the Austrian Federal Ministry for Climate Action, Environment, Energy, Mobility, Innovation and Technology as part of the funding project GZ 2021-0.664.668 (J.S.), and the Austrian Science Fund FWF as part of the funding project 10.55776/EFP5 (S.T.). We are grateful to Jan Hurt for insightful discussions and valuable feedback throughout this study.

## Author contributions

J.S. and S.T. conceived the work. J.S. and A.B. cleaned and prepared the data. J.S. and A.B. wrote the code. J.S., A.B., and S.T. performed the data analysis. J.S., A.B., and S.T. analyzed and interpreted the results. J.S. and S.T. wrote the paper. J.S., A.B., and S.T. contributed to the final manuscript.

## Competing interests

The authors declare no competing interests.
