## [Transparent Peer Review file · Nature Communications]

Using firm-level supply chain networks to measure the speed of the energy transition

Corresponding Author: Professor Stefan Thurner

Version 0:

Reviewer comments:

Reviewer #1

(Remarks to the Author)

Overall, the paper is clearly structured, well-written, and supported by informative figures and tables. However, I identified several critical issues that merit further attention. Due to these fundamental shortcomings, I cannot recommend the paper for acceptance at this stage. In particular, the current version has not yet provided convincing evidence that the chosen methodology is appropriate for answering the research questions posed. My detailed comments are outlined below.

1. The status quo and trend analyses rely on only four reference years, two of which fall within the periods of the COVID-19 pandemic and the energy crisis. Therefore, the results may have limited explanatory power. I understand that the regulatory framework changed in 2020, making earlier data incomparable. Nevertheless, a key question remains: how do the authors ensure that the observed trends are substantially driven by energy transition efforts rather than being primarily a result of these crises?
2. Particularly the second research question “What characteristics distinguish transitioning firms—those with positive decarbonization trends, δ_i , and rates, λ_i ,—from non-transitioning firms?” seems only to be insufficiently answered in the paper. The analysis focuses on energy costs as a key factor, but many other company characteristics likely influence transition behavior. A more comprehensive examination of company-level attributes would be necessary to identify both drivers and barriers to the transition. It may not be sufficient to conclude that the results highlight a lock-in effect because lock-in effects have already been analyzed in existing literature, and this conclusion is not well-founded on the results of this study. While the results may not contradict the presence of a lock-in effect, they seem not to provide causal evidence for it. Moreover, the analysis omits other relevant factors from which such lock-in effects may arise. Given that lock-in phenomena have already been extensively discussed in the literature, the overall contribution of the paper appears limited and may not, in its current form, justify publication as a novel insight.
3. Does the cost analysis consider national or international carbon pricing mechanisms? Since the costs for emitting carbon dioxide are associated with the use of fossil fuels, these costs should be taken into account. Otherwise, the authors should explain why excluding them makes sense.
4. Are the authors referring to direct electrification in comparison to fossil fuel use or the use of low-carbon energy? Particularly when referring to Table II in the text, the electricity share is argued with by the authors, but Table II seems to provide the shares of low-carbon electricity as one category and fossil fuel and conventional electricity use as the other category. Given the pace of transformation in the electricity sector and the efficiency gains from using electricity instead of fossil fuels, I would recommend focusing on direct electrification rather than distinguishing between conventional and low-carbon electricity use. Not distinguishing between two types of electricity also makes sense because the authors derive the carbon dioxide intensity of electricity from the current and expected future Hungarian electricity mix and therefore do not consider the type of electricity specified in company contracts. At the very least, the authors should use consistent terminology to avoid confusion.
5. Other measures to reduce greenhouse gas emissions beyond direct electrification are mentioned only briefly and are not considered in the scenario analysis. The authors should clearly explain why this limitation is appropriate in the context of the scenario analysis and why excluding these other measures is a valid assumption.

6. The policy recommendations section requires further development. In particular, the implications of the trend analysis are presented in vague terms, and the finding that current trends are insufficient to meet emission reduction targets is too general. The authors should derive concrete policy recommendations or strategies to support the uptake of low-carbon energy. For example, the paper could address how policymakers or stakeholders should prioritize subsidies or investments based on the findings. In addition, the authors could explore barriers to the transition beyond the lock-in effect and analyze characteristics that distinguish transitioning from non-transitioning companies.

7. The validation of the approach is currently limited to three NACE groups. I would strongly recommend a more holistic validation by comparing the estimated energy consumption levels with official statistics across all relevant consumption categories. Such a comparison is needed to demonstrate that the method accurately reflects real-world conditions. Eurostat provides data at a sufficiently detailed level.

Some formal and minor language issues are

8. The phrase "decarbonize the electricity grid" in the introduction and discussion may be misleading. A more appropriate expression could be "decarbonize (or defossilize) the electricity sector," which better captures the intended meaning.

9. In Fig. 2, the NACE-1 codes are shown without explanation. As the figure contains considerable unused space, I recommend adding brief labels directly within the figure to clarify what each code represents. In addition, including a table (either in the main text or appendix) that lists all NACE codes used in the paper along with their corresponding sector descriptions would make the analysis more accessible to readers unfamiliar with the coding system.

10. The graphs in Fig. 3 are difficult to interpret, primarily due to the non-intuitive use of a logarithmic scale on the cost share axes. I recommend adopting a more accessible visualization format to improve readability and facilitate interpretation.

11. The use of capital letters for categorizing both elements in the figures and NACE sectors may reduce clarity. This issue is particularly evident in Figure 8 of the Supplementary Information, where capital letters are used for multiple purposes, leading to potential confusion.

(Remarks on code availability)

Reviewer #2

(Remarks to the Author)

This paper offers a novel firm-level evaluation of the energy transition using supply chain data. Overall, it makes significant contribution to the energy transition at a higher-resolution. The paper innovatively used supply-chain data to present a more comprehensive perspective than industries or an individual firm. However, the study has several limitations identified as follows and needs a major revision to fully realize its potential.

Introduction:

1. The introduction outlines a literature gap between broad sectoral analyses and narrow case studies, and the paper aims to bridge these two extremes. However, much of the introduction focuses on data processing and scenario construction rather than articulating the core scientific contribution. I suggest the authors more clearly state the scale of their analysis, the specific subjects (e.g., firms, industries), and how the supply chain network structurally links these actors. This would help clarify the novelty and scientific relevance of the study.

2. Figure 1(a) currently lacks prominence and does not effectively highlight the underlying structure of the supply chain network. It primarily conveys that the dataset includes a large number of firms, but does not illustrate any meaningful topological or sectoral relationships. I recommend that the authors zoom in on specific links and nodes to showcase important interactions within the network.

3. In a supply chain network, firms can simultaneously act as suppliers and consumers. The manuscript should clarify how such dual-role firms are treated in the analysis. This point is important to address, as neglecting the complexity of supply-demand interdependence could make the method less distinguishable from prior literature.

Results:

4. In Figure 3, it is unclear how firms with one of λ and δ equal to zero and the other non-zero are classified. Are such firms considered transitioning, non-transitioning, or excluded? Please clarify this in both the main text and the figure caption to avoid confusion.

5. The observation that "transitioning firms exhibit higher electricity costs while non-transitioning firms have higher fossil costs" seems intuitive and somewhat anticipated. A more meaningful contribution would be to explore additional dimensions such as firm-level cost-benefit trade-offs, business volume, or strategic constraints that influence transition decisions.

Methods:

6. When calculating the low-carbon electricity share, the author implicitly assumes the low-carbon share of the Hungary grid applies uniformly to all domestic electricity suppliers. This assumption could be challenged because the supply chain data do not distinguish renewable from nonrenewable electricity providers but your method suggests providers supply both and clean electricity with a fixed proportion.

7. The authors classify firms into "transitioning" and "non-transitioning" and then used logistic regressions to quantify the relation between energy costs and transition status. This regression, however, is a little simple because only one predictor is used. Given the large panel, more sophisticated models could be applied such as the model that incorporates firm sizes, technological levels, and geography.

8. This article focuses on one country and a short period (2020–2023). This limits the generality of the research. The authors

suggest the method is universal but offer no supporting evidence from other contexts. Moreover, the analysis period includes disruptions from COVID-19 and the energy crisis, which may introduce transient effects rather than reflecting long-term trends. The scenario projections also assume constant energy consumption over time, disregarding potential changes due to new technologies or energy efficiency improvements. I recommend incorporating uncertainty analysis or sensitivity testing to validate the robustness of both methods and conclusions.

9. It is also important to discuss the potential cascading effects along the supply chain, as a firm's decarbonization behavior may influence its suppliers or customers. These interdependencies can generate dynamic, system-wide impacts that shape the overall pace of the energy transition. The paper would contribute significantly by addressing how supply chain structures and relationships may evolve in response to decarbonization pressures.

Policy implication:

10. The discussion section would benefit from more concrete policy recommendations. While the paper correctly highlights the need for targeted incentives to overcome the observed "lock-in" effect, it should go beyond general calls for subsidies. Consider discussing mechanisms such as carbon pricing, emissions standards, or performance-based incentives tailored to firm characteristics and sectoral barriers.

(Remarks on code availability)

The code is complete and could be a technically reusable resource to the community.

Reviewer #3

(Remarks to the Author)

This paper addresses an important and underexplored aspect of the energy transition: the role of firms. While government policy has been the primary driver of decarbonisation efforts, broader participation by individuals and businesses is essential to achieve climate goals. This paper makes a valuable contribution by demonstrating how firm-level data can shed light on businesses' decarbonisation behaviours.

The authors exploit a rich dataset that captures the energy portfolios of 27,000 Hungarian firms in 2020 and 2023. Using this data, they document trends in decarbonisation and identify barriers to progress. Their analysis reveals a worrying picture: at current rates, only 20% of energy consumption is projected to shift to low-carbon electricity by 2050. The paper identifies a "lock-in" effect, whereby firms with higher fossil fuel cost-to-revenue ratios are less likely to transition. Through simulations, the authors show that if laggards were to emulate the efforts of frontrunners within their industries, the share of low-carbon energy could increase to as much as 86%.

The paper is clearly written and methodologically transparent. In my view, its key contribution lies in illustrating how firm-level data can be used to uncover decarbonisation constraints and inform tailored policy interventions.

That said, the paper could be strengthened in several ways:

1. Clarifying the 'lock-in' mechanism

The lock-in explanation is intuitively plausible—firms heavily reliant on fossil fuels may face higher costs or technological barriers to transition. However, the current evidence is based on correlations from logistic regressions. These correlations may reflect underlying firm characteristics (e.g., technology, market power, strategy) rather than causal relationships. Understanding the causal mechanism is critical for designing effective policy interventions. While a full causal analysis may be beyond the scope of this paper, the authors should explicitly acknowledge this limitation. For instance, a future study could employ a difference-in-differences design using exogenous policy variation to assess causal effects. At minimum, the paper should clarify that policy design requires a more rigorous identification of the mechanisms driving high fossil fuel cost shares.

2. Accounting for exogenous shocks

The study period includes two major disruptions: the COVID-19 pandemic and the 2022 energy price shock. These events likely affected firms heterogeneously, even within industries—due to geographic, technological, or organisational differences. While it may not be feasible to control for these shocks in the current analysis, a discussion of their potential effects and how they might bias the results would add valuable context.

3. Assumptions in energy cost conversion

The conversion of financial data into energy units using average energy prices assumes uniform pricing across firms. In reality, larger firms may benefit from long-term contracts or volume discounts, regardless of their fuel mix or technology. This assumption could distort estimates of energy use and transition dynamics. Again, while addressing this empirically may be infeasible, the authors should discuss the implications of this simplifying assumption.

4. Exclusion of self-generated electricity

The omission of self-generated electricity is a potentially significant limitation. Firms investing in on-site renewables or co-generation may have made substantial progress in decarbonisation that the dataset fails to capture. Can the authors provide any data or references on the extent of self-generation in Hungarian industry, and comment on how this exclusion might bias the results?

Minor suggestion:

It would be useful to include a brief discussion on the generalisability of the results. Are there particular features of the Hungarian economy or policy environment that make these findings specific to Hungary? Or could similar patterns be expected in other Central or Eastern European economies?

(Remarks on code availability)

Version 1:

Reviewer comments:

Reviewer #1

(Remarks to the Author)

Dear authors,

Thank you for your comprehensive and thoughtful response to my previous comments. The revised manuscript demonstrates notable improvements in methodological transparency, analytical scope, and the discussion of limitations.

Nevertheless, I remain somewhat ambivalent. While many of my initial concerns—particularly regarding the short observation period, the reliance on trend analysis, which generally seems not to be the ideal method for analyzing scenarios of the future energy system, and the limited novelty of the results—have been addressed or mitigated, they have not been fully resolved. The number of reference years remains low, and the inherent constraints of trend-based analysis persist.

On balance, I nevertheless believe that the improvements, including the open discussion of the study's limitations, justify publication. I therefore recommend acceptance of the revised manuscript.

(Remarks on code availability)

Reviewer #2

(Remarks to the Author)

The author has made a thorough revisions of the manuscript following the reviewers' suggestions. The revised manuscript showcases a comprehensive analysis of the energy transition of firms. I have no additional comments for revision.

(Remarks on code availability)

Reviewer #3

(Remarks to the Author)

I thank the authors for their responsive revisions. They have satisfactorily addressed many of my earlier concerns. However, I still have several objections, which can likely be resolved through careful redrafting rather than substantive changes.

1. Lack of Causality

I welcome the change in language, which now suggests that the results are consistent with a lock-in effect. However, without establishing causality, the policy implications remain limited. Sound policymaking depends on identifying mechanisms, not correlations. I would be satisfied with a clear disclaimer acknowledging this limitation, and even more so with an indication that future work may attempt to establish causal links—perhaps through a well-designed quasi-experimental approach.

2. Energy Cost Conversion

I appreciate the authors' explanation that energy use was converted to cost using average prices across seven electricity consumption bands and six for gas. Nonetheless, my concern remains that applying uniform prices within each band may distort measures of transition dynamics. Firms within the same band can face very different prices and opportunities for decarbonization.

For example, a large university and a medium-sized industrial enterprise operating 24 hours a day may both fall within the same consumption band, yet their load profiles and access to renewable energy options differ markedly. The university's predominantly daytime load and larger footprint create more scope for self-generation or solar procurement, implying different effective prices.

I recognize the value of this first attempt to use detailed firm-level data, but the manuscript should explicitly acknowledge these limitations and discuss the likely implications. Which types of firms or industries might face lower effective prices, and how could this affect observed decarbonization trends?

3. Exclusion of Self-Generation

The discussion of this limitation (lines 572–586) remains unconvincing. The claim that “the share of commercial and industrial PV in Hungary is probably lower than the PV average of 30%” limits aggregate omission bias is not persuasive.

For instance, even if the national average is only 20%, a heavy concentration of PV in specific industries could significantly alter decarbonisation trends within those sectors. The paper should provide a more robust discussion of the potential impact of excluding self-generation, or at least acknowledge that this omission may bias results for some industries.

4. Readability and Accuracy

The paper would benefit from a careful language and consistency review. A few illustrative issues:

- Line 133: still refers to annual prices.
- Lines 187–190: sentence structure is awkward and unclear.

(Remarks on code availability)

Response to reviewer #1

Overall, the paper is clearly structured, well-written, and supported by informative figures and tables. However, I identified several critical issues that merit further attention. Due to these fundamental shortcomings, I cannot recommend the paper for acceptance at this stage. In particular, the current version has not yet provided convincing evidence that the chosen methodology is appropriate for answering the research questions posed. My detailed comments are outlined below.

1. The status quo and trend analyses rely on only four reference years, two of which fall within the periods of the COVID-19 pandemic and the energy crisis. Therefore, the results may have limited explanatory power. I understand that the regulatory framework changed in 2020, making earlier data incomparable. Nevertheless, a key question remains: how do the authors ensure that the observed trends are substantially driven by energy transition efforts rather than being primarily a result of these crises?

We thank the reviewer for this very important comment, which has prompted us to substantially revise our methodology. We agree that the inclusion of years affected by the COVID-19 pandemic and the energy crisis could potentially limit the explanatory power of the results, and we have therefore revised our analysis to ensure that the observed trends reflect underlying transition dynamics rather than temporary crisis effects. Specifically, we implemented the following changes:

1. We extended the observation period by including an additional year, 2024, made possible by newly available data from the Hungarian central bank (adapted throughout the text).
2. We switched to semi-annual aggregation of the predominantly monthly firm-to-firm transaction data, which allows a closer alignment with semi-annual energy price data reported by EUROSTAT. To account for possible seasonality, we then aggregate back to annual estimates of energy consumption and derive the results for λ_i and δ_i from the resulting low-carbon shares of five years (see Methods section *Conversion of monetary inputs into energy consumption via energy prices*, p. 12).
3. We replaced the ordinary least squares (OLS) estimator with a robust regression estimator based on the Huber loss function, which down-weights the influence of large residuals. This improves our ability to capture long-term decarbonization trends while reducing the impact of short-term fluctuations caused by COVID-19, the energy crisis, or firm-specific shocks (see Methods, p. 13; Supplementary Methods and Supplementary Fig. 6).

In addition, we tested the robustness of our energy scenarios (see Supplementary Discussion *Uncertainty analysis of the energy scenarios*, pp. 4-5, and Supplementary Fig. 15). We use a leave-one-year-out procedure, similar to bootstrapping, re-estimating λ_i and δ_i while sequentially omitting each year from 2020–2024. Then we re-run each scenario with the re-estimated coefficients. For each scenario, this produces five runs in total and allows us to construct an uncertainty envelope around the main run (all years 2020–2024), defined by the minimum and maximum low-carbon shares across runs. The results show that both linear and exponential scenarios are generally stable. The only notable deviation occurs when 2020 is omitted, leading to faster projected growth in the exponential scenarios. This is intuitive,

since 2020 has the highest aggregate electricity share in our firm sample; excluding it means slopes are re-estimated only from 2021–2024, resulting in more positive trends. Despite this sensitivity, the overall consistency across runs underscores that our results are robust and primarily capture structural transition dynamics rather than short-term crisis effects.

3. Does the cost analysis consider national or international carbon pricing mechanisms? Since the costs for emitting carbon dioxide are associated with the use of fossil fuels, these costs should be taken into account. Otherwise, the authors should explain why excluding them makes sense.

We thank the reviewer for raising this important point. Carefully considering this comment, we decided to exclude firms participating in the European Union Emissions Trading System (ETS), as their energy costs are affected by additional expenditures for carbon credits, which cannot be accounted for in our analysis of firm characteristics. Our dataset is anonymized, allowing us to identify whether a company participates in the ETS but not to determine which specific firms they resemble. Therefore, estimating their energy-related carbon credit costs would require individual treatment that is not feasible within the scope of this study. For this reason, we adopt a conservative approach and exclude ETS firms from the sample, amounting to 201 firms in total. This approach is outlined in the revised Methods section *Firm sample construction* (pp. 11-12). This exclusion affects the total energy coverage and the coverage of manufacturing firms, since ETS firms tend to be large consumers of energy of that sector. However, we consider it more important to remain consistent with our analysis of firm-level characteristics and to shed light on the energy transition behaviour of firms not currently covered by an emissions trading scheme, which could not be studied comprehensively before.

4. Are the authors referring to direct electrification in comparison to fossil fuel use or the use of low-carbon energy? Particularly when referring to Table II in the text, the electricity share is argued with by the authors, but Table II seems to provide the shares of low-carbon electricity as one category and fossil fuel and conventional electricity use as the other category. Given the pace of transformation in the electricity sector and the efficiency gains from using electricity instead of fossil fuels, I would recommend focusing on direct electrification rather than distinguishing between conventional and low-carbon electricity use. Not distinguishing between two types of electricity also makes sense because the authors derive the carbon dioxide intensity of electricity from the current and expected future Hungarian electricity mix and therefore do not consider the type of electricity specified in company contracts. At the very least, the authors should use consistent terminology to avoid confusion.

We thank the reviewer for raising this point. We are indeed referring to direct electrification, and after revisiting the manuscript we recognized that the used terminology may have caused confusion. We ensured that the concept of the low-carbon share is clearly introduced in the Introduction and used consistently throughout the manuscript. After careful consideration, we decided to keep the distinction between low-carbon and conventional electricity, but we now make this explicit in the revised figures and text. In particular, we have updated Fig. 1b and its caption to clarify that we do not distinguish between renewable and conventional electricity providers within the supply chain network, but instead apply the low-carbon share of the Hungarian electricity mix only at a later stage (see Fig. 1, p. 2). In addition, we revised Fig. 4 to illustrate how the decarbonization of the Hungarian electricity sector factors into our energy scenarios by displaying both the share of low-carbon electricity and the share of

conventional electricity (see Fig. 4, p. 8). We agree that the pace of transformation in the electricity sector will likely lead to a fully low-carbon electricity sector in Hungary in the not so distant future, as also reflected in our forecast (Supplementary Methods *Electricity mix of Hungary and forecast until 2050*; p. 1, and Supplementary Fig. 1).

The specific carbon intensity of the electricity purchased by firms cannot be derived from our supply chain data. However, the Hungarian electricity market is highly concentrated, with one dominant firm group (MVM) dominating the market, and it is therefore reasonable to assume that the average firm procures electricity with the carbon intensity approximating the overall national mix (see also IEA country report on Hungary, 2022: <https://www.iea.org/reports/hungary-2022>). While this assumption may not hold for some highly ambitious firms that actively source renewable electricity, it provides a valid approximation for the majority of firms purchasing from the main provider. We now explicitly state this in the revised Methods section *Calculating the low-carbon share $l_i(t)$* (pp. 12-13).

5. Other measures to reduce greenhouse gas emissions beyond direct electrification are mentioned only briefly and are not considered in the scenario analysis. The authors should clearly explain why this limitation is appropriate in the context of the scenario analysis and why excluding these other measures is a valid assumption.

We thank the reviewer for this important remark. We agree that additional decarbonization pathways, such as CCUS, synthetic fuels, or hydrogen, play a crucial role, particularly in energy-intensive sectors such as cement and chemicals. In this study, however, we concentrate on direct electrification for two main reasons. First, it is widely recognized as the primary decarbonization pathway for most sectors, as highlighted by several studies cited in the Introduction [IPCC AR6 Report on Climate Mitigation; Mercure et al., 2021] (see Introduction, p. 1). Second, our analysis is limited to energy consumption categories that can be inferred from firm-level industry codes, namely conventional sources such as oil, gas, and electricity. If product-level information becomes available in future studies, the framework could be extended to incorporate additional pathways, such as hydrogen, CCUS, and synthetic fuels.

We have added corresponding comments at the end of the Results section (p. 6) and in the Discussion section (p. 10).

6. The policy recommendations section requires further development. In particular, the implications of the trend analysis are presented in vague terms, and the finding that current trends are insufficient to meet emission reduction targets is too general. The authors should derive concrete policy recommendations or strategies to support the uptake of low-carbon energy. For example, the paper could address how policymakers or stakeholders should prioritize subsidies or investments based on the findings. In addition, the authors could explore barriers to the transition beyond the lock-in effect and analyze characteristics that distinguish transitioning from non-transitioning companies.

We thank the reviewer for this important remark. We have carefully revised the policy recommendation section in the revised Discussion and incorporated additional insights from our extended analysis of firm characteristics. We have also rewritten the section on the energy scenario analysis, placing less emphasis on compliance with international climate targets and focusing instead on the concrete implications that can be drawn from the results. First, current trends do not allow for a substantial reduction in fossil fuel use in the near

future. Second, in virtually all fine-grained NACE 4-digit categories, we identify firms that increase their shares of low-carbon electricity, suggesting that no fundamental technological barriers hinder the transition. Third, this observation indicates that a substantial increase in the uptake of low-carbon electricity could be achieved if firms were to follow the behaviour of industry frontrunners.

Building on these findings, we now provide more concrete implications for how policies could support and accelerate movement onto such transition pathways. Lowering investment barriers could be achieved through subsidies for electrification technologies or through instruments such as state-backed loans on favorable terms to help firms manage the high upfront costs of replacing fossil-based capital. Since high-revenue firms appear less likely to transition, policy instruments that reduce the attractiveness of continued fossil fuel use may be necessary. Stronger carbon pricing is one broad option, but because our study provides a tool to identify frontrunners and laggards within fine-grained industry segments, policy-makers could also design more targeted measures, such as sectoral emission standards or performance-based incentives that reward greener firms through mechanisms like tax breaks or preferential access to public procurement. Finally, given the positive correlation between high total energy consumption and transition behaviour, large energy consumers that demonstrate a credible decarbonization pathway should therefore also benefit from favorable treatment (see Discussion, pp. 9–11).

7. The validation of the approach is currently limited to three NACE groups. I would strongly recommend a more holistic validation by comparing the estimated energy consumption levels with official statistics across all relevant consumption categories. Such a comparison is needed to demonstrate that the method accurately reflects real-world conditions. Eurostat provides data at a sufficiently detailed level.

We thank the reviewer for this important comment. We agree that validation should be strengthened and have expanded our analysis accordingly. In the revised manuscript, we include many more NACE categories (see Supplementary Discussion *Comparison of firm sample to sectoral energy consumption data*, p. 3 and Supplementary Figs. 9-12, pp. 10-13). We now compare our estimates with official statistics for the NACE 1-digit sector F – Construction. We further incorporate a new data source on industrial energy use in Hungary, which enables us to validate our firm sample against official statistics at the NACE 2-digit level of the manufacturing sector. Our results show that the firm sample in general reflects the relative importance of oil, gas, and electricity consumption for most sectors. However, gas appears to be overrepresented, while electricity is somewhat underrepresented, leading to lower aggregate electricity shares in our firm sample compared to official statistics. This discrepancy may be explained by the fact that manufacturing firms also procure electricity through channels such as the spot market, which our NACE-based method cannot capture. In addition, our analysis relies on a restricted sample of firms, so it remains inherently uncertain to what extent the gap in electricity use relative to official statistics reflects firms not included in our dataset.

In response, we have adopted more cautious language throughout the manuscript, particularly when describing the energy scenarios. We also note that 2022 appears to be an exceptional year in our sample, further motivating the use of a robust estimator to derive energy trends.

Some formal and minor language issues are 8. The phrase “decarbonize the electricity grid” in the introduction and discussion may be misleading. A more

appropriate expression could be “decarbonize (or defossilize) the electricity sector,” which better captures the intended meaning.

We thank the reviewer for this helpful clarification. We have adopted the suggested terminology in the abstract and in the main text (see e.g. Abstract, Introduction, p. 1).

9. In Fig. 2, the NACE-1 codes are shown without explanation. As the figure contains considerable unused space, I recommend adding brief labels directly within the figure to clarify what each code represents. In addition, including a table (either in the main text or appendix) that lists all NACE codes used in the paper along with their corresponding sector descriptions would make the analysis more accessible to readers unfamiliar with the coding system.

We thank the reviewer for this helpful comment. We have revised Fig. 2 to include a legend with the NACE 1-digit codes and descriptions. As only abbreviated descriptions could be added due to space limitations, we also provide a full table of NACE 1-digit codes and their descriptions in Supplementary Tab. V, p. 18. In addition, a table of NACE 2-digit codes for the manufacturing sector is now included in Supplementary Tab. IV.

10. The graphs in Fig. 3 are difficult to interpret, primarily due to the non-intuitive use of a logarithmic scale on the cost share axes. I recommend adopting a more accessible visualization format to improve readability and facilitate interpretation.

We thank the reviewer for this valuable remark. We have replaced Fig. 3 with a forest plot of the estimated coefficients, which we believe provides a clearer and more accessible representation of the differences in firm characteristics between transitioning and non-transitioning firms (see Fig. 3, p. 5).

11. The use of capital letters for categorizing both elements in the figures and NACE sectors may reduce clarity. This issue is particularly evident in Fig. 8 of the Supplementary Information, where capital letters are used for multiple purposes, leading to potential confusion.

We thank the reviewer for this helpful remark. In response, we have revised the manuscript to use lower-case letters for subfigures throughout, thereby avoiding potential confusion with capital letters used for NACE categories.

Response to reviewer #2

This paper offers a novel firm-level evaluation of the energy transition using supply chain data. Overall, it makes significant contribution to the energy transition at a higher-resolution. The paper innovatively used supply-chain data to present a more comprehensive perspective than industries or an individual firm. However, the study has several limitations identified as follows and needs a major revision to fully realize its potential.

Introduction:

1. The introduction outlines a literature gap between broad sectoral analyses and narrow case studies, and the paper aims to bridge these two extremes. However, much of the introduction focuses on data processing and scenario construction rather than articulating the core scientific contribution. I suggest the authors more clearly state the scale of their analysis, the specific subjects (e.g., firms, industries), and how the supply chain network structurally links these actors. This would help clarify the novelty and scientific relevance of the study.

We thank the reviewer for this insightful comment. In the revised manuscript, we more clearly emphasize our key scientific contribution: showing how increasingly available supply chain network data derived from value-added tax and electronic invoices can be used to analyze the dynamics of the energy transition comprehensively at the firm level (see p. 2).

We agree that the second part of the introduction contains a rather detailed discussion of methods. Since we view our methodological approach as a central scientific contribution, we believe it is important to introduce it already in the introduction, while keeping the main technical details in the Methods section. Our analysis primarily focuses on the direct consumption of energy, which is why the supply chain network itself was not the main focus of the introduction. However, following the reviewer’s suggestion, we now explicitly discuss the transitioning behaviour of firms in relation to that of their suppliers and customers in Supplementary Discussion *Influence of suppliers and customers on transitioning behaviour*, p. 4. See also Supplementary Figs. 13, 14 (pp. 13-14).

2. **Figure 1(a) currently lacks prominence and does not effectively highlight the underlying structure of the supply chain network. It primarily conveys that the dataset includes a large number of firms, but does not illustrate any meaningful topological or sectoral relationships. I recommend that the authors zoom in on specific links and nodes to showcase important interactions within the network.**

We thank the reviewer for this valuable comment. We agree that Fig. 1a does not provide structural insight into the supply chain network; its intended purpose is to illustrate that our analysis builds on comprehensive supply chain data. We conceptually “zoom in” on the network in Fig. 1b to highlight the distinction between energy-providing and energy-consuming firms, since these connections are most relevant to our analysis of firms’ transition behaviour. The structure of the Hungarian supply chain network itself has already been studied extensively (e.g., Borsos et al., 2020; Diem et al., 2022), both of which are referenced in our introduction. This insightful comment has, however, led us to extend our analysis by examining whether the transition behaviour of firms is aligned with that of their suppliers and customers (see Supplementary Discussion *Influence of suppliers and customers on transitioning behaviour*, p. 4. See also Supplementary Figs. 13, 14 (pp. 13-14). A more comprehensive study of the role of supply chain topology in the energy transition is beyond the scope of the present paper, but our work provides a foundation for exploring this promising research

direction in the future.

3. In a supply chain network, firms can simultaneously act as suppliers and consumers. The manuscript should clarify how such dual-role firms are treated in the analysis. This point is important to address, as neglecting the complexity of supply-demand interdependence could make the method less distinguishable from prior literature.

We thank the reviewer for this valuable comment. This is exactly the reason why we have omitted energy-providing firms themselves from our analyses, since they would be both energy providers and consumers, obfuscating the change in energy demand. Our sample is carefully constructed to only contain energy-consuming firms. This insightful comment led us to exclude an additional NACE 4-digit category “H52.2.1 – Service activities incidental to land transportation” from our analysis, as firms from this category might be involved in liquefaction of gas for transportation purposes which likely means their purchases are not linked to consumption of gas directly (see Methods *Firm sample construction*, p. 11).

Results:

4. In Fig. 3, it is unclear how firms with one of λ and δ equal to zero and the other non-zero are classified. Are such firms considered transitioning, non-transitioning, or excluded? Please clarify this in both the main text and the figure caption to avoid confusion.

We thank the reviewer for this important point, which was not explicitly addressed before. In our analysis, we classify only firms for which both λ_i and δ_i are positive as transitioning firms. Firms for which either of the two is negative are considered non-transitioning. This represents a conservative classification of (non-)transitioning firms. We now clarify this classification in the caption of the revised Fig. 3 (p. 5) as well as in the main text (p. 3) and in the Methods section *Quantifying the relationship between firm characteristics and transition status* (pp. 13-14).

5. The observation that “transitioning firms exhibit higher electricity costs while non-transitioning firms have higher fossil costs” seems intuitive and somewhat anticipated. A more meaningful contribution would be to explore additional dimensions such as firm-level cost-benefit trade-offs, business volume, or strategic constraints that influence transition decisions.

We thank the reviewer for this important comment. In response, and also following the advice of other reviewers, we have substantially revised the section on firm characteristics associated with the transition behaviour. Rather than limiting the analysis to comparisons of electricity and fossil energy costs, we now perform a multivariate logistic regression across NACE 1-digit sectors (see Fig. 3, p. 5 and Table I, p. 7). This approach quantifies how several firm characteristics (average fossil energy cost share, average electricity cost share, average revenue, average employment, and average total energy consumption) simultaneously affect the odds of transitioning. The resulting adjusted odds ratios (AOR) represent the change in transition odds associated with a 10% increase in each characteristic (see Results section *Characteristics of transitioning firms*, pp. 4-6).

This extended analysis provides a more nuanced picture of the characteristics associated with transition. While the discrepancy between fossil and electricity cost shares remains, we additionally find that firms with higher revenue are less likely to transition, whereas firms with higher energy consumption are more likely. We discuss these findings and their policy

implications in the revised Discussion section (pp. 9-10).

Methods:

6. When calculating the low-carbon electricity share, the author implicitly assumes the low-carbon share of the Hungary grid applies uniformly to all domestic electricity suppliers. This assumption could be challenged because the supply chain data do not distinguish renewable from nonrenewable electricity providers but your method suggests providers supply both and clean electricity with a fixed proportion.

We thank the reviewer for raising this point, which helped us to be more explicit in our methodology. We agree that this assumption can be challenged, since our data do not allow us to distinguish between renewable and non-renewable electricity providers. However, the Hungarian electricity market is highly concentrated, with one firm group (MVM) as the dominant provider (see also the IEA country report on Hungary, 2022: <https://www.iea.org/reports/hungary-2022>). Thus, while the assumption may not hold for highly ambitious firms that actively source renewable electricity, it is a reasonable approximation for the average firm purchasing from the most prevalent provider. We now explicitly state this in the Methods section *Calculating the low-carbon share $l_i(t)$* (pp. 12-13). In addition, we have revised Fig. 1b as well as the caption of Fig. 1 to ensure that it does not suggest we distinguish between renewable and conventional electricity providers within the supply chain network, but rather that we apply the low-carbon share of the Hungarian electricity mix only at a later stage (see Fig. 1, p. 2).

7. The authors classify firms into “transitioning” and “non-transitioning” and then used logistic regressions to quantify the relation between energy costs and transition status. This regression, however, is a little simple because only one predictor is used. Given the large panel, more sophisticated models could be applied such as the model that incorporates firm sizes, technological levels, and geography.

We thank the reviewer for this insightful comment which, together with comment 5 and the advice from other reviewers, led us to completely revise our analysis of firm characteristics in relation to transition status. While we do not have access to data on technological levels or geography (which would indeed be valuable dimensions to explore), we now incorporate several measures of firm size (revenue, employment, and total energy consumption) into the logistic regression analysis. Moreover, we replaced the earlier univariate regressions with a multivariate logistic regression, allowing us to simultaneously control for the influence of multiple firm characteristics. To avoid collinearity, we excluded the total energy cost share, as it is highly correlated with fossil and electricity cost shares and would obscure their individual effects in the regression. In addition, we log-transform variables to reduce skewness and rely on multi-year averages rather than single-year values to obtain more robust estimates. We believe these changes significantly improve the robustness and explanatory power of our analysis. The revised results are presented in the Results section *Characteristics of transitioning firms* (pp. 4-6), in Fig. 3 (p. 5), and Table I (p. 7), with their implications discussed in the revised Discussion section (pp. 9-10).

8. This article focuses on one country and a short period (2020–2023). This limits the generality of the research. The authors suggest the method is universal but offer no supporting evidence from other contexts. Moreover, the analysis period includes disruptions from COVID-19 and the energy crisis, which may

introduce transient effects rather than reflecting long-term trends. The scenario projections also assume constant energy consumption over time, disregarding potential changes due to new technologies or energy efficiency improvements. I recommend incorporating uncertainty analysis or sensitivity testing to validate the robustness of both methods and conclusions.

We thank the reviewer for this important comment, which together with the advice from other reviewers led us to substantially revise our analysis on several key aspects. To arrive at more robust estimates of the decarbonization trends λ_i and δ_i and rates, we introduced the following changes:

1. We extended the observation period by including an additional year, 2024, made possible through newly available data from the Hungarian central bank (adapted throughout the text).
2. We switched to semi-annual aggregation of the predominantly monthly firm-to-firm transaction data, which allowed us to more accurately match the semi-annual energy prices reported by EUROSTAT. To counteract seasonality, the estimated energy consumption was then aggregated back to annual values (see Methods section *Conversion of monetary inputs into energy consumption via energy prices*, p. 12).
3. We replaced the ordinary least squares (OLS) estimator with a robust regression estimator based on the Huber loss function, which down-weights the influence of large residuals by combining quadratic and linear loss. This approach reduces the impact of potential fluctuations caused by COVID-19, the energy crisis, or firm-specific disruptions, thereby improving our ability to capture long-term trends (see Methods *Measuring the speed of the energy transition at the firm level*, p. 13; Supplementary Methods *OLS vs. robust estimation* (p. 2); Supplementary Fig. 6).

We now additionally assess the robustness of the energy scenarios using a leave-one-year-out procedure, similar to bootstrapping. In this approach, λ_i and δ_i are re-estimated while sequentially omitting each year from 2020-2024. Then we re-run each scenario with the re-estimated coefficients. For each scenario, this produces five runs in total and allows us to construct an uncertainty envelope around the main run (all years 2020-2024), defined by the minimum and maximum low-carbon shares across runs. The results indicate that both linear and exponential scenarios are generally stable. The only notable deviation arises when 2020 is omitted, which leads to faster projected growth of the low-carbon share in the exponential scenarios. This outcome is intuitive, as 2020 shows the highest electricity share in our sample; excluding it means slopes are re-estimated only from 2021–2024, resulting in more positive trends. Despite this sensitivity, the strong consistency across runs underscores the robustness of our results (see Supplementary Discussion *Uncertainty analysis of the energy scenarios*, pp. 4-5; Supplementary Fig. 15, p. 15).

We carefully considered to also test the assumption of extended average energy consumption, but chose not to add this additional layer of complexity, since our scenarios are intended primarily for illustration. Their purpose is to show that current firm-level decarbonization trends are insufficient to meaningfully reduce energy consumption, whereas substantial reductions could be achieved if firms followed the frontrunners in their industries. We state this now more clearly in the revised Discussion section (p. 9-10).

9. It is also important to discuss the potential cascading effects along the supply chain, as a firm’s decarbonization behavior may influence its suppliers or cus-

tomers. These interdependencies can generate dynamic, system-wide impacts that shape the overall pace of the energy transition. The paper would contribute significantly by addressing how supply chain structures and relationships may evolve in response to decarbonization pressures.

We thank the reviewer for this important comment, which prompted us to investigate the potential cascading effects along the supply chain. We discuss this in Supplementary Discussion *Influence of suppliers and customers on transitioning behaviour* (p. 4) and Supplementary Figs. 13,14 (pp. 13-14). We also included a reference in the Discussion of the main text (p. 9-10).

In this new analysis, we examine the relationship between a firm’s decarbonization behaviour and that of its suppliers and customers. First, we correlate each firm’s decarbonization trend, δ_i , with the mean decarbonization trend of its suppliers and customers, δ_j , across tiers 1–3. We find negative correlations with suppliers, positive correlations with direct customers (tiers 1 and 2), and negative correlations with customers at tier 3. Second, we correlate each firm’s transition status with the share of its suppliers and customers that are transitioning. Here we again find negative correlations for suppliers and positive correlations for direct customers. These results suggest that firms’ transition behaviour is more closely aligned with their customers than with their suppliers, which could be interpreted as pressure from customers to decarbonize.

We emphasize, however, that these results should be interpreted with caution. Our analysis only captures partner behaviour within the firm sample, and the number of observed partners grows rapidly across tiers, which may influence the correlations. We therefore present this section as an exploratory analysis. It provides initial indications that supply-chain interdependencies may shape firms’ transition decisions and serves as motivation for future research to more rigorously investigate the cascading effects of decarbonization behaviour along supply chains.

Policy implication:

10. The discussion section would benefit from more concrete policy recommendations. While the paper correctly highlights the need for targeted incentives to overcome the observed “lock-in” effect, it should go beyond general calls for subsidies. Consider discussing mechanisms such as carbon pricing, emissions standards, or performance-based incentives tailored to firm characteristics and sectoral barriers.

We thank the reviewer for this important remark. In response, we have carefully revised the policy recommendation section in the Discussion to provide more concrete and differentiated policy implications. Building on the extended analysis of firm characteristics and the revised energy scenario analysis, we now highlight several mechanisms that could support firms in overcoming barriers to transition.

First, we emphasize that lowering investment barriers remains a central priority. This could be achieved not only through subsidies for electrification technologies, but also via instruments such as state-backed loans on favorable terms, which can help firms manage the high upfront costs of replacing fossil-based capital. Second, since high-revenue firms appear less likely to transition, policy instruments that reduce the attractiveness of continued fossil fuel use may be required. Stronger carbon pricing is one broad option. In addition, because our study provides a tool to identify frontrunners and laggards within fine-grained industry segments, policymakers could also design more targeted measures, such as sectoral emissions

standards or performance-based incentives that reward greener firms through mechanisms like tax breaks or preferential access to public procurement. Finally, given the positive correlation between high total energy consumption and transition behaviour, we recommend that large energy consumers demonstrating a credible decarbonization pathway should similarly receive preferential treatment, reflecting their potential leverage in reducing emissions.

These revisions provide a more concrete set of policy options tailored to the firm-level and sectoral barriers identified in our analysis (see Discussion section, pp. 9-10).

Reviewer #2 (Remarks on code availability):

The code is complete and could be a technically reusable resource to the community.

The code provided with this paper has been adapted in accordance with our methodology and deposited in the accompanying repository.

Response to reviewer #3

This paper addresses an important and underexplored aspect of the energy transition: the role of firms. While government policy has been the primary driver of decarbonisation efforts, broader participation by individuals and businesses is essential to achieve climate goals. This paper makes a valuable contribution by demonstrating how firm-level data can shed light on businesses' decarbonisation behaviours.

The authors exploit a rich dataset that captures the energy portfolios of 27,000 Hungarian firms in 2020 and 2023. Using this data, they document trends in decarbonisation and identify barriers to progress. Their analysis reveals a worrying picture: at current rates, only 20% of energy consumption is projected to shift to low-carbon electricity by 2050. The paper identifies a “lock-in” effect, whereby firms with higher fossil fuel cost-to-revenue ratios are less likely to transition. Through simulations, the authors show that if laggards were to emulate the efforts of frontrunners within their industries, the share of low-carbon energy could increase to as much as 86%.

The paper is clearly written and methodologically transparent. In my view, its key contribution lies in illustrating how firm-level data can be used to uncover decarbonisation constraints and inform tailored policy interventions.

That said, the paper could be strengthened in several ways:

1. Clarifying the ‘lock-in’ mechanism. The lock-in explanation is intuitively plausible—firms heavily reliant on fossil fuels may face higher costs or technological barriers to transition. However, the current evidence is based on correlations from logistic regressions. These correlations may reflect underlying firm characteristics (e.g., technology, market power, strategy) rather than causal relationships. Understanding the causal mechanism is critical for designing effective policy interventions. While a full causal analysis may be beyond the scope of this paper, the authors should explicitly acknowledge this limitation. For instance, a future study could employ a difference-in-differences design using exogenous policy variation to assess causal effects. At minimum, the paper should clarify that policy design requires a more rigorous identification of the mechanisms driving high fossil fuel cost shares.

We thank the reviewer for this important comment. We agree that our initial language around the potential lock-in effect was too strong, since we cannot establish causality and we now use more careful wording, stating that the results from the logistic regression are consistent with a lock-in effect (see Abstract; Results, *Characteristics of transitioning firms*, pp. 4-6; Discussion, pp. 9-10).

In response to this and related comments, we have substantially revised our logistic regression analysis of firm characteristics. Specifically, we now incorporate several measures of firm size (revenue, employment, and total energy consumption) to control for potential confounding factors. We also replaced the earlier univariate regressions with a multivariate logistic regression framework, which allows us to account for multiple firm characteristics simultaneously. To avoid collinearity, we exclude the total energy cost share, which is highly correlated with fossil and electricity cost shares. Furthermore, we log-transform skewed variables and rely on averages across years to obtain more robust estimates rather than depending on a single year. We believe that these changes substantially improve the robustness and explanatory

power of the analysis. The revised results are presented in the Results section *Characteristics of transitioning firms* (pp. 4-6), in Fig. 3 (p. 5), and Table I (p. 7), with their implications discussed in the Discussion section (pp. 9-10).

2. Accounting for exogenous shocks. The study period includes two major disruptions: the COVID-19 pandemic and the 2022 energy price shock. These events likely affected firms heterogeneously, even within industries—due to geographic, technological, or organisational differences. While it may not be feasible to control for these shocks in the current analysis, a discussion of their potential effects and how they might bias the results would add valuable context.

We thank the reviewer for this important comment. We agree that the COVID-19 pandemic and the 2022 energy price shock were major disruptions that likely affected firms in heterogeneous ways, and similar concerns have also been raised by other reviewers. While it is not possible to precisely identify the effects of these crises on individual firms, we have substantially revised our analysis to mitigate their potential influence and to provide more robust estimates of firm-level decarbonization trends λ_i and rates δ_i .

1. We extended the observation period by including 2024, made possible through newly available data from the Hungarian central bank (adapted throughout the text).
2. We switched to semi-annual aggregation of the predominantly monthly firm-to-firm transaction data, which allowed us to more accurately match the semi-annual energy prices reported by EUROSTAT. To counteract seasonality, the estimated energy consumption was then aggregated back to annual values (see Methods section *Conversion of monetary inputs into energy consumption via energy prices*, p. 12).
3. We replaced the ordinary least squares (OLS) estimator with a robust regression estimator based on the Huber loss function, which down-weights the influence of large residuals by combining quadratic and linear loss. This reduces the impact of fluctuations caused by COVID-19, the energy crisis, or firm-specific disruptions and improves our ability to capture long-term trends (see Methods *Measuring the speed of the energy transition at the firm level*, p. 13; Supplementary Methods *OLS vs. robust estimation* (p. 2); Supplementary Fig. 6.)

In addition, we now assess the robustness of the energy scenarios using a leave-one-year-out procedure, similar to bootstrapping. In this approach, λ_i and δ_i are re-estimated while sequentially omitting each year from 2020–2024. Then we re-run each scenario with the re-estimated coefficients. For each scenario, this produces five runs in total and allows us to construct an uncertainty envelope around the main run (all years 2020–2024), defined by the minimum and maximum low-carbon shares across runs. The results indicate that both linear and exponential scenarios are generally stable. The only notable deviation arises when 2020 is omitted, which leads to faster projected growth of the low-carbon share in the exponential scenarios. This outcome is intuitive, as 2020 shows the highest electricity share in our sample; excluding it means slopes are re-estimated only from 2021–2024, resulting in more positive trends. Despite this sensitivity, the strong consistency across runs underscores the robustness of our results (see Supplementary Discussion *Uncertainty analysis of the energy scenarios*, pp. 4-5; Supplementary Fig. 15, p. 15).

3. Assumptions in energy cost conversion. The conversion of financial data into energy units using average energy prices assumes uniform pricing across firms. In reality, larger firms may benefit from long-term contracts or volume

discounts, regardless of their fuel mix or technology. This assumption could distort estimates of energy use and transition dynamics. Again, while addressing this empirically may be infeasible, the authors should discuss the implications of this simplifying assumption.

We thank the reviewer for this insightful comment, which indeed raises important concerns about the assumptions in our energy cost conversion methodology. As described in the Methods section *Conversion of monetary inputs into energy consumption via energy prices* (p. 12), we do not assume a single uniform price across firms. Instead, we use heterogeneous electricity and gas prices from EUROSTAT, stratified by consumption bands. The semi-annual price evolution shown in Supplementary Figs. 2-3 (p. 6) demonstrates that larger consumers experience price changes earlier, which may indicate that these firms also participate in spot market energy trading. However, we acknowledge the reviewer’s point that the energy prices used in our analysis represent averages within the respective consumption bands, which remains a limitation. We have added a corresponding note in the Supplementary Discussion *Electricity and gas price evolution in Hungary* (p.2).

At the same time, our conversion procedure has been significantly improved by aggregating purchase data on a semi-annual basis, allowing us to apply the reported energy prices more directly.

Nonetheless, our approach has limitations: we only observe purchases from traditional energy providers, meaning that electricity or gas obtained through other channels is not captured. As discussed in Supplementary Discussion *Comparison of firm sample to sectoral energy consumption data* (pp. 3-4) and shown in Supplementary Figs. 9-12 (pp. 10-13), electricity use appears underestimated for certain manufacturing sectors, which may indicate that firms procure additional electricity through sources (like the spot market) not identifiable with our method. We now highlight this limitation more clearly in the revised Discussion section (pp. 9-10).

4. Exclusion of self-generated electricity

The omission of self-generated electricity is a potentially significant limitation. Firms investing in on-site renewables or co-generation may have made substantial progress in decarbonisation that the dataset fails to capture. Can the authors provide any data or references on the extent of self-generation in Hungarian industry, and comment on how this exclusion might bias the results?

We thank the reviewer for this important comment. We agree that the omission of self-generated electricity is a relevant limitation and have carefully considered how this affects our results. While we could not identify sources reporting self-generation or firm-level PV installations directly, data from SolarPower Europe indicate that commercial and industrial solar accounts for roughly one third of total installed capacity. By contrast, the IEA Photovoltaic Power Systems Programme (IEA PVPS) highlights in its report *Trends in Photovoltaic Applications 2024* that utility-scale projects are the main driver of PV capacity growth in Hungary. Taken together, these sources suggest that while excluding self-generation may lead us to underestimate progress for some individual firms, the aggregate bias is likely limited, as the share of commercial and industrial PV capacity in Hungary is probably lower than the EU average of 30%. We include these two sources and discuss this limitation in the revised Discussion section (pp. 9-10).

Minor suggestion:

It would be useful to include a brief discussion on the generalisability of the

results. Are there particular features of the Hungarian economy or policy environment that make these findings specific to Hungary? Or could similar patterns be expected in other Central or Eastern European economies?

We thank the reviewer for raising this important point about generalisability. We now include an additional paragraph in the revised Discussion (p. 9-10) addressing this issue. Several characteristics of the Hungarian economy, including its continued reliance on fossil fuels, the presence of energy-intensive industries, the expansion of nuclear power, and the rapid uptake of solar PV, are shared with other Central and Eastern European economies, like Slovakia. While the quantitative results are context-specific, some qualitative insights are likely to carry over, such as the structural role of energy costs in shaping transition behaviour and the observation that virtually all fine-grained industry sectors include both frontrunners that rapidly adopt low-carbon energy sources and laggards that continue to expand their fossil fuel use.

Response to reviewer #1

Dear authors,

Thank you for your comprehensive and thoughtful response to my previous comments. The revised manuscript demonstrates notable improvements in methodological transparency, analytical scope, and the discussion of limitations.

Nevertheless, I remain somewhat ambivalent. While many of my initial concerns—particularly regarding the short observation period, the reliance on trend analysis, which generally seems not to be the ideal method for analyzing scenarios of the future energy system, and the limited novelty of the results—have been addressed or mitigated, they have not been fully resolved. The number of reference years remains low, and the inherent constraints of trend-based analysis persist.

On balance, I nevertheless believe that the improvements, including the open discussion of the study's limitations, justify publication. I therefore recommend acceptance of the revised manuscript.

We thank the reviewer for the careful consideration and for the recommendation for acceptance. In response to comments from another reviewer, we have further expanded the discussion of the limitations of our study.

In particular, we clarify that the correlational nature of our findings regarding firm characteristics and transition behaviour does not permit causal interpretation, and that any policy implications should therefore be interpreted with caution. We added a new section to our Supplementary Discussion to report an unsuccessful attempt to implement the Granger non-causality test by Dumitrescu and Hurlin (2012) and to discuss how future studies could attempt to identify the causal mechanisms behind the observed correlations (p. 5). We also elaborate on the limitations of our energy price data: while prices vary by consumer size, firms of similar size may have substantially different load profiles and thus face different effective prices, which may result in both under- and overestimation (Discussion, pp. 9-10). In addition, we have added another section to the Supplementary Discussion that provides a detailed calculation of the potential bias arising from the omission of self-consumed solar PV generation (pp. 5-6). Although commercial self-consumption is growing, this bias is likely modest, particularly for large industrial consumers.

We hope that these additional clarifications further strengthen the manuscript and help alleviate any remaining concerns.

Response to reviewer #2

The author has made a thorough revisions of the manuscript following the reviewers' suggestions. The revised manuscript showcases a comprehensive analysis of the energy transition of firms. I have no additional comments for revision.

We thank the reviewer for the careful evaluation of the manuscript and for the positive assessment of our revisions. We appreciate the acknowledgement of our efforts to strengthen the analysis and address the earlier concerns. We are grateful for the reviewer's time and support.

Response to reviewer #3

I thank the authors for their responsive revisions. They have satisfactorily addressed many of my earlier concerns. However, I still have several objections, which can likely be resolved through careful redrafting rather than substantive changes.

1. Lack of Causality I welcome the change in language, which now suggests that the results are consistent with a lock-in effect. However, without establishing causality, the policy implications remain limited. Sound policymaking depends on identifying mechanisms, not correlations. I would be satisfied with a clear disclaimer acknowledging this limitation, and even more so with an indication that future work may attempt to establish causal links—perhaps through a well-designed quasi-experimental approach.

We thank the reviewer for acknowledging our revisions and for the important remark regarding the need to further clarify the limitations related to causality. In response, we have added a clear disclaimer to the Discussion section of the revised manuscript (p. 9) and have further softened the language of the policy implications. We now explicitly state that our empirical strategy does not establish causal relationships, emphasize that the policy implications should therefore be interpreted with appropriate caution, and note that future research could identify causal mechanisms using quasi-experimental or other causal inference approaches.

To expand on this point, we have added a dedicated section to the Supplementary Discussion (p. 5). There, we describe our unsuccessful attempt to apply the heterogeneous panel Granger non-causality test of Dumitrescu and Hurlin (2012) to explore whether firm characteristics exhibit temporal precedence with respect to subsequent rises or declines in their low-carbon energy share. Although this method relaxes some requirements of traditional Granger tests, our panel remains too short for meaningful inference: the individual regressions become unstable, and the resulting statistics are not interpretable. We note that future research with a longer time series, or with suitable quasi-experimental variation, would be needed to robustly identify causal relationships.

2. Energy Cost Conversion I appreciate the authors' explanation that energy use was converted to cost using average prices across seven electricity consumption bands and six for gas. Nonetheless, my concern remains that applying uniform prices within each band may distort measures of transition dynamics. Firms within the same band can face very different prices and opportunities for decarbonization.

For example, a large university and a medium-sized industrial enterprise operating 24 hours a day may both fall within the same consumption band, yet their load profiles and access to renewable energy options differ markedly. The university's predominantly daytime load and larger footprint create more scope for self-generation or solar procurement, implying different effective prices.

I recognize the value of this first attempt to use detailed firm-level data, but the manuscript should explicitly acknowledge these limitations and discuss the likely implications. Which types of firms or industries might face lower effective prices, and how could this affect observed decarbonization trends?

We thank the reviewer for raising this important point. In response, we have added an additional paragraph to the Discussion section explicitly acknowledging that applying uniform

prices within consumption bands masks heterogeneity in effective firm-level prices due to differences in load profiles, contractual arrangements, and opportunities for on-site generation or renewable procurement. We briefly outline which firms are likely to face lower or higher effective prices, such as daytime-oriented organizations versus continuously operating industrial firms (p.10).

As a result, we may underestimate the low-carbon share of firms with daytime-oriented load profiles and overestimate it for firms with more even, industrial load profiles. If this bias is systematic, the general trends in our fitted models are unlikely to change substantially, although the intercept may shift. Therefore, the forecasted aggregate low-carbon shares in the scenario analysis may be somewhat overstated, given the high energy use of industrial consumers.

We highlight that our results should be interpreted with this limitation in mind, and future work could incorporate more granular pricing data or load-profile information to better capture these firm-level cost differences.

3. Exclusion of Self-Generation The discussion of this limitation (lines 572–586) remains unconvincing. The claim that “the share of commercial and industrial PV in Hungary is probably lower than the PV average of 30%” limits aggregate omission bias is not persuasive.

For instance, even if the national average is only 20%, a heavy concentration of PV in specific industries could significantly alter decarbonisation trends within those sectors. The paper should provide a more robust discussion of the potential impact of excluding self-generation, or at least acknowledge that this omission may bias results for some industries. Discussion: Acknowledge this (industrial firms might be able to cover a higher share from self-generation, as their premises allow for more solar installation, etc.)

We thank the reviewer for this valuable comment. In response, we have added a whole new section to the Supplementary Discussion (pp. 5-6) and a paragraph to the Discussion section of the main text (p.10). We draw on several data sources, including Hungary’s transmission system operator MAVIR, the Hungarian Energy and Utilities Regulatory Authority MEKH, and ENTSO-E, to derive estimates of commercially self-consumed PV electricity. We estimate that commercially self-consumed PV electricity increased from roughly 0.6 TWh in 2020 to about 2.5 TWh in 2024, corresponding to around 27% of Hungary’s total annual PV output and covering an estimated 2.3% of total commercial electricity consumption in 2020 and about 8.8% in 2024. We note that these figures are approximations, but they suggest that, while self-generation has grown substantially, its aggregate magnitude remains modest relative to total commercial consumption, particularly for large industrial users with very high electricity demand.

At the same time, we now explicitly acknowledge in the Discussion section (p.5) that the impact is likely heterogeneous across sectors. Firms with predominantly daytime loads, such as service-sector organizations, may have comparatively higher PV self-consumption shares relative to their total demand, whereas large industrial firms typically consume much larger volumes, making the relative contribution of PV self-generation smaller. We also note that some industrial firms may have invested in large-scale PV systems for self-consumption that could lead to material deviations, but such cases cannot be identified in our data and therefore remain a limitation.

Accordingly, we clarify that our estimated low-carbon electricity shares may be understated

for certain service-sector firms and that this limitation should be kept in mind when interpreting industry-level patterns. We also highlight that future work could address this limitation by integrating firm-level PV installation or load-profile data if it becomes available.

4. Readability and Accuracy The paper would benefit from a careful language and consistency review. A few illustrative issues:

- **Line 133:** still refers to annual prices.
- **Lines 187–190:** sentence structure is awkward and unclear.

We thank the reviewer for pointing out these issues. We have carefully reviewed the manuscript for language clarity and consistency and corrected the specific instances mentioned. We have also conducted a broader language edit to improve readability throughout the manuscript.